# Evolution of the tuberculin skin test reveals generalisable *Mtb*-reactive T cell metaclones

Carolin T. Turner [1,18], Andreas Tiffeau-Mayer [1,2,18], Joshua Rosenheim [1,14], Aneesh Chandran[1,15], Rishika Saxena [1], Ping Zhang [3], Jana Jiang [1,4], Michelle Berkeley[1], Flora Pang[1], Imran Uddin [1,16], Gayathri Nageswaran [1], Suzanne Byrne [1], Akshay Karthikeyan[1], Werner Smidt [5], Paul Ogongo [6,17], Rachel Byng-Maddick [1], Santino Capocci[7,8], Marc Lipman[9,10], Heinke Kunst [11], Stefan Lozewicz [12], Veron Ramsuran [5,13], Gabriele Pollara [1], Julian C. Knight [3], Alasdair Leslie [1,6], Benjamin M. Chain [1] & Mahdad Noursadeghi [1,8] ✉

T cells contribute to immune protection and pathogenesis in tuberculosis, but measurements of polyclonal responses have failed to resolve correlates of outcome. We report the temporal evaluation of the human in vivo clonal repertoire of *Mycobacterium tuberculosis* (*Mtb*)-reactive T cell responses, by T cell receptor (TCR) sequencing at the site of the tuberculin skin test, as a model for a standardised antigenic challenge. Initial non-selective recruitment of T cells is followed by enrichment of *Mtb*-reactive clones arising from oligo-clonal T cell proliferation. We introduce a modular computational pipeline, Metaclonotypist, to sensitively cluster distinct TCRs with shared epitope specificity, which we apply here to establish a catalogue of public *Mtb*-reactive HLA-restricted T cell metaclones. Although most in vivo *Mtb*-reactive T cells are private, 10 metaclones were sufficient to identify *Mtb*-T cell reactivity across our study population (N≥128), indicating striking population level immunodominance of specific TCR-peptide interactions that may inform patient stratification and vaccine development.

*Mycobacterium tuberculosis* (*Mtb*) remains the commonest microbial cause of death worldwide, but most incident infections do not progress to tuberculosis (TB) disease[1,2]. Immunodeficiency is associated with increased risk of disease, indicating a role for protective immunity. However, disease is mediated by immunopathology, associated with a failure to restrict *Mtb* growth. Better understanding of the immune correlates of protection and pathogenesis remain global research priorities to inform novel approaches to disease-risk

[1]Division of Infection and Immunity, University College London, London, UK. [2]Institute for the Physics of Living Systems, University College London, London, UK. [3]Centre for Human Genetics, University of Oxford, Oxford, UK. [4]Eye Center, Medical Center, Faculty of Medicine, University of Freiburg, Freiburg, Germany. [5]Centre for the AIDS Programme of Research in South Africa, Durban, South Africa. [6]Africa Health Research Institute, Durban, South Africa. [7]Department of Respiratory Medicine, University College London Hospitals NHS Foundation Trust, London, UK. [8]North Central London TB service, Whittington Health NHS Trust, London, UK. [9]Department of Respiratory Medicine, Royal Free London NHS Foundation Trust, London, UK. [10]UCL Respiratory, Division of Medicine, University College London, London, UK. [11]Queen Mary & Barts Health Tuberculosis Centre, Blizard Institute, Faculty of Medicine & Dentistry, Queen Mary University London, London, UK. [12]Department of Respiratory Medicine, North Middlesex University Hospital, London, UK. [13]School of Medicine, College of Health Sciences, University of KwaZulu-Natal, Durban, South Africa. [14]Present address: Department of Immunology and Microbiology, University of Copenhagen, Copenhagen, Denmark. [15]Present address: Institute of Dentistry, Queen Mary University London, London, UK. [16]Present address: Cancer Institute, University College London, London, UK. [17]Present address: Division of Experimental Medicine, University of California San Francisco, San Francisco, CA, USA. [18]These authors contributed equally: Carolin T. Turner, Andreas Tiffeau-Mayer. ✉e-mail: m.noursadeghi@ucl.ac.uk

stratification in *Mtb* infected people, vaccine development and evaluation, and identification of targets for host-directed therapies.

T cells are essential for protective immunity to TB. They are thought to augment bacterial restriction within intracellular niches such as macrophages[3,4]. T cell-mediated protection against TB is evident in increased disease risk associated with genetic deficiencies of IL-12 and IFNγ signalling[5], T cell depletion in people living with HIV[6], and experimental T cell depletion in non-human primates[7,8]. Yet, frequency of circulating *Mtb*-reactive T cells, and limited analysis of their functional attributes (cytokine production or cytolytic degranulation) do not predict natural or vaccine-inducible protective immunity in humans[4]. We have also reported evidence for T cell-mediated pathogenesis in TB, illustrated by enrichment of IL-17 producing T cells in people with disease compared to those who have controlled infection[9], disease triggered by checkpoint inhibitor therapies that increase effector T cell function[10] and direct stimulation of *Mtb*-growth by IFNγ[11].

T cells exist as clonal populations identified by their T cell receptor (TCR), most commonly composed of αβ heterodimers produced by imprecise somatic gene recombination during T cell development and responsible for signalling T cell activation following recognition of antigen bound to MHC molecules. *Mtb* proteome-wide studies have identified immunodominant protein antigens[12–14]. To date, use of whole protein or pooled peptide antigens to quantify *Mtb*-reactive T cells has not resolved correlates of protection and pathogenesis, potentially because they measure polyclonal responses in which responses to distinct peptide-MHC targets have differential effects on outcome. TCR sequencing enables an antigen-agnostic approach to resolve clonal T cell responses. This has provided proof of concept for potential correlates of outcome[15], but the generalisability of these findings is not known.

Importantly, studies of human T cell biology in TB have relied heavily on investigation of *Mtb* reactive T cells from blood samples that are limited by sampling depth, because they contain <0.001% of the T cell clonal repertoire of an individual[16], only a small fraction of which is *Mtb* reactive. Alternatively, investigation of T cells from the site of disease, such as bronchoalveolar lavage specimens or tissue biopsies, can enrich for the antigen-specific cells of interest, but is confounded by the chronicity of infection and pathological processes. We have addressed these limitations by profiling immune responses at the site of the tuberculin skin test (TST)[9,17–19]. Tuberculin is a standardised clinical grade preparation of purified protein derivative (PPD) from *Mtb*. Inflammatory induration at the site of the TST after 2−3 days has been used extensively as a classical model of delayed type hypersensitivity dependent on T cell priming, and therefore a measure of T cell memory for *Mtb* antigens contained in PPD. We have previously used this model to quantify T cell recruitment and function, and to reveal exaggerated IL17 activity associated with active TB disease, which could not be detected in blood[9]. The clonal repertoire of the T cell response to *Mtb* challenge in vivo has not previously been systematically evaluated. The extent to which these responses converge onto dominant T cell clones, and whether these are generalisable or idiosyncratic within a population are not known.

We addressed these questions by TCR sequencing of biopsies from the site of the TST in a cohort of 223 individuals. This approach provided us with a sensitive, unbiased quantitation of T cell clones recruited and expanded in response to a standardised in vivo challenge. To evaluate convergence of the response to immunodominant epitopes, we developed Metaclonotypist, a modular bioinformatics pipeline for the grouping of TCR sequences based on sequence similarity. Using this pipeline, we discover a dominant immune response to TB driven by highly public HLA-associated TCR metaclonotypes, which we expect to be a valuable resource for future biomarker discovery and reverse epitope discovery efforts in tuberculosis.

## Results

### Transcriptome-wide evaluation of maturation of the immune response to TST from day 2 to day 7

Inflammatory induration in the TST is maximal at 2−3 days, but previous flow cytometric evaluation of T cells at the site of the TST reported maximal accumulation of antigen-specific T cell responses at 7 days[20]. Therefore, we investigated the evolution of the T cell response by bulk RNAseq and TCRseq in day 2 and day 7 TSTs (Supplementary Fig. 1). We recruited healthy volunteers with evidence of peripheral blood *Mtb*-reactive T cells identified during occupational health screening, TB index case contact screening or recent migrant screening, to undergo a TST in each arm (Table 1). The TST site was sampled on day 2 at one site and on day 7 at the contralateral site. Genome-wide TST-response transcriptomes at day 2 and day 7 were defined by differential gene expression compared to transcriptomes from the site of control saline injections performed in a separate set of volunteers. The TST-response transcriptomes at each time point were used to infer activity of immune response pathways at the level of cytokines, receptors, kinases and transcription factors. Both day 2 and day 7 TST transcriptomes showed activation of a comparable repertoire of canonical immune signalling pathways (Supplementary Fig. 2A, B). We next identified gene expression modules associated with individual upstream regulators, which were significantly upregulated in integrated data from day 2 and day 7 TST-response transcriptomes. We found that most module expression decreased between day 2 and day 7 (Supplementary Fig. 2C) consistent with homeostatic resolution of inflammatory changes. A small number of modules showed higher expression at day 7. These were all identified as being regulated by transcription factors known to be involved in cell cycle regulation (Supplementary Fig. 2D).

Direct comparison of day 2 and day 7 TST-response transcriptomes also revealed differences in expression at the level of individual genes (Supplementary Fig. 2E). Pathway enrichment analysis of the differentially expressed genes indicated increased cell cycle and mitotic activity in the day 7 TST (Supplementary Fig. 2F), suggestive of increased cell proliferation at this time point. Therefore, we tested the hypothesis that the evolution of transcriptional changes between the day 2 and day 7 TSTs reflected T cell proliferation by quantifying the correlation between independently derived gene expression modules for cellular proliferation and selected T cell and non-T cell subsets. By comparison to day 2 profiles, the transcriptomes from day 7 TSTs showed significantly higher expression of the modules for cell proliferation (Fig. 1A), pan-T cell, CD4 T cells and NK cells, but not modules

**Table 1 | Participant overview for RNA sequencing**

| Characteristic | saline *N* = 33[a] | Day 2 TST *N* = 216[a] | Day 7 TST *N* = 158[a] |
|---|---|---|---|
| Sex | | | |
| Female | 19 (58%) | 115 (53%) | 85 (54%) |
| Male | 14 (42%) | 101 (47%) | 73 (46%) |
| Age | 28 (21, 39) | 34 (29, 43) | 35 (29, 45) |
| Unknown | 1 | 1 | 1 |
| Ethnicity | | | |
| African | 5 (16%) | 64 (30%) | 48 (31%) |
| American | 7 (22%) | 3 (1.4%) | 2 (1.3%) |
| East Asian | 2 (6.3%) | 40 (19%) | 27 (17%) |
| European | 13 (41%) | 64 (30%) | 44 (28%) |
| Mixed | 1 (3.1%) | 1 (0.5%) | 1 (0.6%) |
| South Asian | 4 (13%) | 40 (19%) | 34 (22%) |
| Unknown | 1 | 4 | 2 |

[a]*n* (%); Median (Q1, Q3).

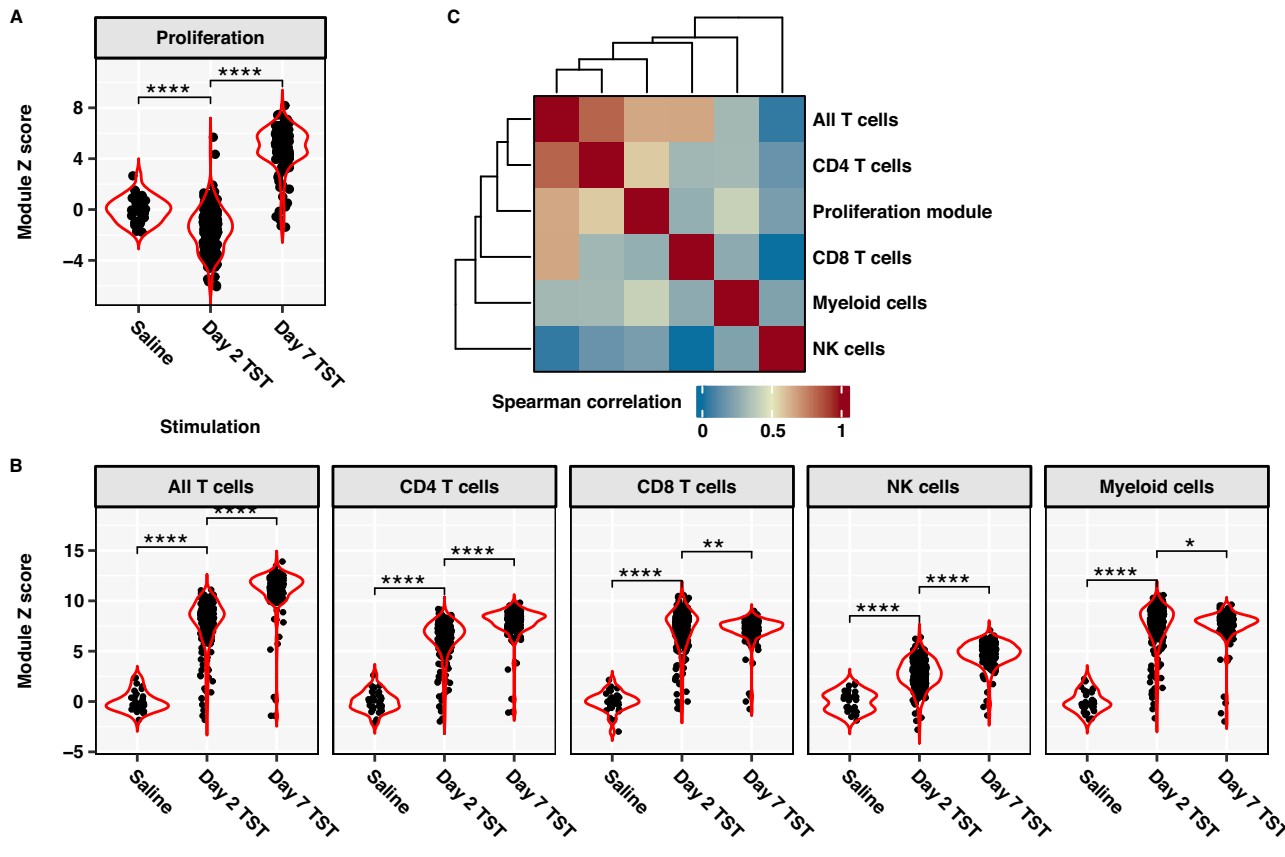

**Fig. 1 | Proliferation response in Day 7 TST is correlated with CD4 T cell gene signature.** Expression of cellular proliferation (**A**) and cell type-specific (**B**) modules in bulk RNA sequencing data from saline-injected control skin, Day 2 and Day 7 TSTs, shown as Z-score scaled TPM expression using saline samples as control group ($n = 33$ saline, $n = 216$ Day 2 TST, $n = 158$ Day 7 TST). Unpaired, two-sided Wilcoxon test with multiple testing correction: * FDR < 0.05, ** FDR < 0.01, **** FDR < 0.0001. **C** Heatmap of Spearman correlation matrix between gene signatures in Day 7 TSTs. Dendrogram depicts average linkage clustering of correlation coefficients.

for CD8 T cells or myeloid cells (Fig. 1B). The cell proliferation module showed greatest correlation to the pan-T cell and CD4 T cell modules (Fig. 1C). Clinical induration at TST sites on day 2 was positively correlated with multiple cell types at day 2, but only T cell accumulation at day 7, as measured by cell-type specific gene expression of modules (Supplementary Table 1).

### Evolution of reduced T cell clonal diversity in the TST

To study the nature of the T cell proliferation response revealed by our transcriptional analysis, we tracked the temporal evolution of the T cell clonal repertoire in the TST by TCRα and TCRβ sequencing of bulk RNA from day 2 and day 7 TSTs (Table 2 and Supplementary Fig. 1). We compared TCR repertoire diversity metrics to those of unstimulated peripheral blood samples from the same population of study participants (Fig. 2A). We display metrics for the TCR β-chain, which is more diverse and informative about TCR antigen specificity[21], but found concordant results for metrics calculated on TCR α-chain repertoires (Supplementary Fig. 3A). The median number of total β-chain TCRs obtained was 267,057 (range 113–566,403) for day 7 TSTs; 53,756 (range 2596–100,381) for day 2 TSTs; and 66,452 (range 18,266–119,984) for peripheral blood samples. Since repertoire diversity metrics are affected by sequencing depth (Supplementary Fig. 3B, C), we excluded samples with very small repertoires and down-sampled repertoires to the same size ($n = 16{,}000$ total TCRs) prior to this analysis. Compared to blood, the day 2 TST repertoire had an increased frequency of TCR sequences with >1 copy and a greater inequality of clone sizes as measured by Gini index, indicative of

### Table 2 | Participant overview for TCR sequencing

| Characteristic | Blood $N = 20^a$ | PBMC $N = 12^a$ | Day 2 TST $N = 17^a$ | Day 7 TST $N = 165^a$ |
|---|---|---|---|---|
| Sex | | | | |
| Female | 9 (45%) | 3 (25%) | 6 (35%) | 88 (53%) |
| Male | 11 (55%) | 9 (75%) | 11 (65%) | 77 (47%) |
| Age | 33 (30, 36) | 34 (30, 37) | 33 (30, 36) | 35 (29, 44) |
| Unknown | 0 | 0 | 0 | 1 |
| Ethnicity | | | | |
| African | 2 (11%) | 2 (18%) | 2 (13%) | 50 (31%) |
| American | 1 (5.3%) | 1 (9.1%) | 1 (6.3%) | 3 (1.9%) |
| East Asian | 0 (0%) | 0 (0%) | 0 (0%) | 27 (17%) |
| European | 15 (79%) | 7 (64%) | 12 (75%) | 48 (30%) |
| Mixed | 0 (0%) | 0 (0%) | 0 (0%) | 1 (0.6%) |
| South Asian | 1 (5.3%) | 1 (9.1%) | 1 (6.3%) | 33 (20%) |
| Unknown | 1 | 1 | 1 | 3 |

$^a n$ (%); Median (Q1, Q3)

recruitment of expanded memory T cell clones. Correspondingly, the number of unique TCR clones (Richness) was reduced compared to blood. In contrast, Shannon diversity did not differ significantly from blood at day 2 and Simpson diversity even slightly increased, indicative of limited clonal dominance at this timepoint. Day 7 TSTs showed still higher proportions of expanded TCR clones, further reduced richness

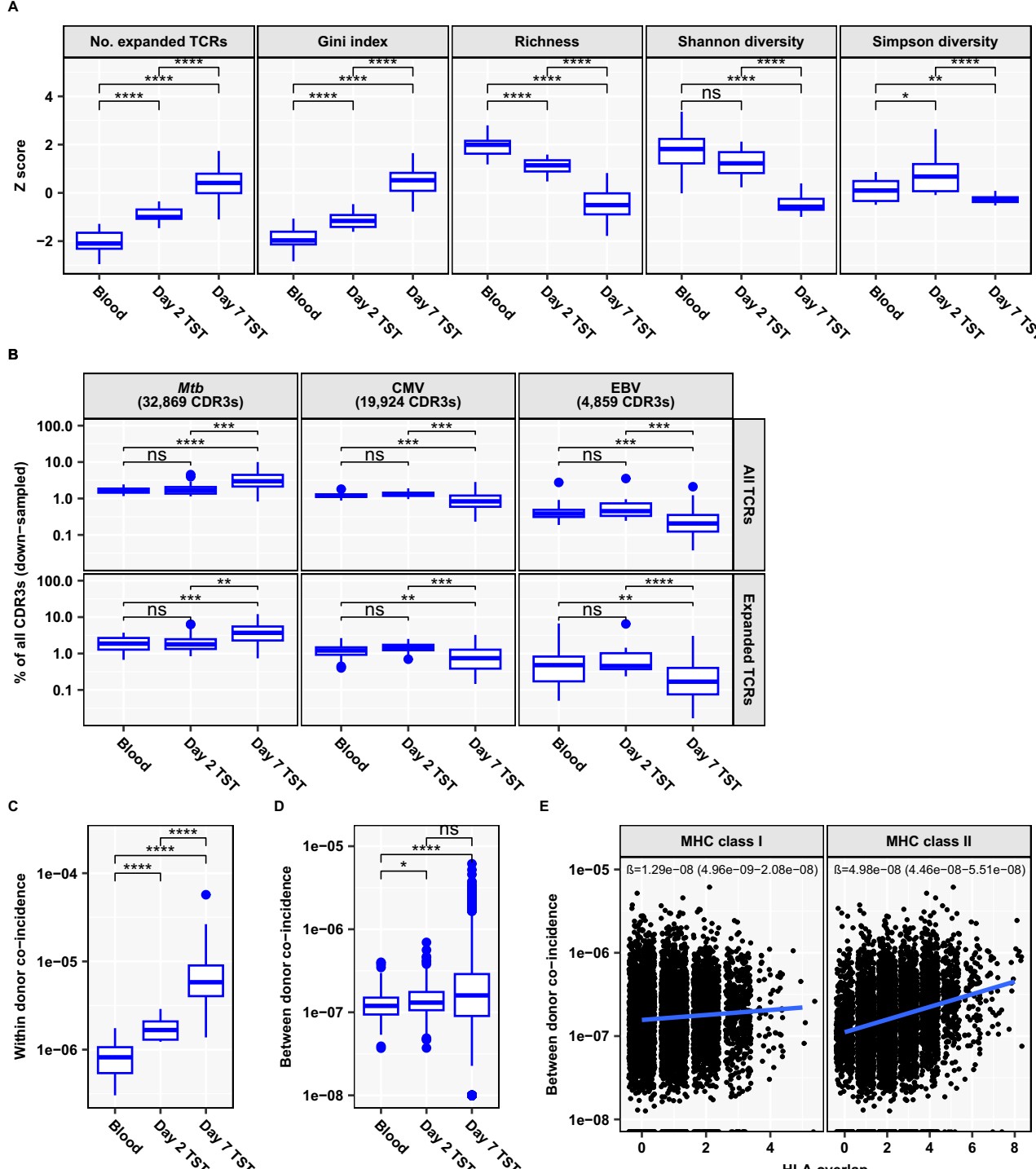

**Fig. 2 | Functional restriction of the TCR repertoire in Day 7 TST yet limited inter-individual TCR sharing.** Individual β-chain bulk TCR repertoires (*n* = 20 Blood, *n* = 16 Day 2 TST, *n* = 128 Day 7 TST) were down-sampled to 16,000 TCRs. Boxplots in (**A**–**D**) depict median and inter-quartile range (IQR), with outlier data points (more than 1.5*IQR beyond the box hinges) shown as dots. Statistical significance was assessed with unpaired, two-sided Wilcoxon tests and corrected for multiple testing (ns FDR > 0.05, * FDR < 0.05, ** FDR < 0.01, **** FDR < 0.0001). **A** TCRβ repertoire diversity metrics, shown as Z-score values scaled across all samples. No. expanded TCRs = number of TCR sequences present more than once. **B** Abundance of published antigen-reactive CDR3 sequences (specific for *Mtb*, CMV or EBV; collated from VDJdb and McPAS databases as well as Musvosvi et al.[15]),

shown as percentage of all TCRs or of all expanded TCRs (present more than once). The number of distinct published antigen-reactive CDR3s available to assess enrichment of antigen reactivity in blood and TST samples is indicated. **C** Within-donor convergence of distinct clones as identified by nucleotide sequence identity onto identical amino acid sequences. **D** Cross-donor TCR convergence, calculated between any two individuals, resulting in *n* = 190 (Blood), *n* = 120 (Day 2 TST) and *n* = 8128 (Day 7 TST) pairwise comparisons. **E** Cross-donor TCR convergence in Day 7 TSTs, stratified by the number of class I or class II HLA alleles shared between any two individuals. Each dot represents a pairwise comparison (*n* = 8128). The linear regression line is shown in blue, with regression coefficient (slope β) and its 95% confidence interval indicated.

and Shannon diversity, and were characterised by the emergence of clonal dominance in the TCR repertoire, measured as decreased Simpson diversity and increased Gini index. Taken together, these findings suggest that the selection of T cell clones recruited to the day 2 TST is not particularly stringent, but repertoires evolve towards oligoclonality as a result of selective CD4 T cell proliferation and clonal expansion.

## Day 7 TST is highly enriched for expanded Mtb-reactive T cell clones

Since inflammatory responses in the TST are dependent on Mtb-reactive T cells, we tested the hypothesis that the day 2 and day 7 TST TCR repertoires are selectively enriched for Mtb-reactive T cells compared to blood samples from the same individuals. We tested this hypothesis first by evaluating enrichment of previously described Mtb-reactive CDR3 sequences. We compared the TST CDR3 β-chain sequences to published CDR3 sequences derived from T cells with known antigen reactivity, including 32,869 reported Mtb-reactive TCRs identified by virtue of pMHC specificity or upregulation of T cell activation markers on ex vivo stimulation with Mtb[15,22,23]. Compared to blood, we found no enrichment for Mtb-reactive CDR3s among all TCRs or all expanded TCRs in the day 2 TST (Fig. 2B and Supplementary Fig. 4). This indicates that the early inflammatory response likely reflects the circulating frequency of Mtb-specific T cells in individuals with prior memory, rather than preferential recruitment of specific clones. However, there was statistically significant enrichment of published Mtb-reactive CDR3 sequences in day 7 TSTs compared to blood and to day 2 TSTs. As a comparison, we similarly calculated the enrichment of published CMV or EBV-reactive CDR3 sequences. Day 7 TSTs showed a statistically significant reduction in the relative frequencies of both CMV and EBV-reactive CDR3 sequences compared to blood and day 2 TSTs, consistent with larger clonal expansions of Mtb-reactive sequences. These clonal expansions were not explained by donor-unrestricted T cell responses to Mtb, since day 7 TSTs showed a significant reduction in the relative frequencies of TCR α sequences that match the gene usage of MAIT or iNKT cells, compared to day 2 TSTs and/or blood (Supplementary Fig. 5).

Next, we reasoned that antigen-driven selection of T cell responses in the TST would lead to increased functional convergence of TCR clones onto common CDR3 amino acid sequences[24]. This convergent sequence evolution was clearly evident within the repertoires of individual participants, which showed a progressive increase from blood to day 2 and then day 7 TST in coincidence probabilities (Fig. 2C and Supplementary Fig. 6A–C). Inter-individual analysis showed a more modest increase in average coincidence probability across day 2 TSTs from pairs of donors compared to blood, and no further significant increase in average coincidence probability among pairs of day 7 TSTs (Fig. 2D and Supplementary Fig. 6D–F). Interestingly, the variance of between donor coincidences increased substantially at day 7, compatible with differential skewing of the Mtb-reactive T cell repertoires among different individuals driven by diversity in MHC-restricted antigen presentation to T cells. To further test this hypothesis, we analysed how probabilities of inter-individual coincidence in day 7 TST TCR sequences depend on HLA-allele sharing between pairs of individuals (Fig. 2E and Supplementary Fig. 6G–I). We found that pairs of individuals sharing multiple MHC class II or class I alleles had substantially more similar day 7 TST repertoires. The dependence of repertoire overlap on HLA similarity was fourfold stronger with MHC class II, consistent with a predominantly CD4 T cell response in the TST.

In view of the potential for inter-individual diversity of the Mtb-reactive TCR repertoire, we reasoned that evaluation of day 7 TSTs using published Mtb-reactive CDR3 sequences may substantially underestimate the enrichment of Mtb-reactive T cell clones because this analysis is inherently restricted to public TCRs. Therefore, we

experimentally validated Mtb-reactive TCRs at the level of individual participants. We sequenced TCRs of peripheral blood mononuclear cells (PBMC) from a sample of the study population, following ex vivo stimulation for 6 days with PPD, or tetanus toxoid (TT) as antigen control, and selected all the CDR3s which expanded eightfold or more in the PPD-stimulated cultures but not the control unstimulated cultures[25,26]. The expanded PPD-reactive CDR3 sequences showed limited publicity among the sub-sampled study participants (Fig. 3A, B and Supplementary Fig. 7A, B). We therefore refer to these PPD-reactive CDR3 sequences as private. We then looked for overlap between the CDR3 sequences of in vitro expanded T cells and the TST repertoires from the same participant. There was no enrichment of ex vivo PPD-expanded CDR3s in the day 2 TST repertoires compared to unstimulated blood (Fig. 3C, D and Supplementary Fig. 6C–H). Both in blood and the day 2 TST, ex vivo PPD-expanded CDR3s were present in significantly greater proportion amongst TCR sequences with a count >1 (Fig. 3C, D and Supplementary Fig. 7C–H), suggesting that the PPD-reactive CDR3s were predominantly expanded, as would be expected for memory T cells[26]. We found significantly greater enrichment of ex vivo PPD-expanded CDR3s in day 7 TSTs than in unstimulated blood repertoires or in day 2 TST repertoires. This pattern remained the same whether overlap was calculated for total CDR3 sequences or for unique CDR3 sequences (Fig. 3C, D and Supplementary Fig. 7C–H). This enrichment of private PPD-reactive CDR3s in day 7 TSTs was further increased among the most expanded TCRs. No similar enrichment of ex-vivo TT-expanded CDR3s was observed. Additionally, the odds ratio (OR) for the overlap with ex vivo PPD-reactive CDR3s was substantially greater among CDR3s which significantly expanded between day 2 and day 7 TSTs, compared to non-expanded CDR3 sequences (Supplementary Fig. 8). Taken together, these results indicate that antigen non-specific accumulation of T cells in the day 2 TST is largely replaced by expanded Mtb-reactive T cell clones by day 7 post-TST.

## Identification of Mtb-reactive metaclones in the day 7 TST

Identification of T cell metaclones, defined by similar but non-identical CDR3 sequences, which share specificity for the same peptide-MHC, can address the limitations of interindividual TCR sequence diversity and enable antigen-agnostic identification of generalisable T cell responses to specific pMHC targets[24,27,28]. The identification of metaclones involves clustering of TCRs by sequence similarity, followed by a test for HLA-association across a cohort (Fig. 4A). A number of approaches to clustering of TCR sequences have been proposed. Among these, the GLIPH2 algorithm has already been used to identify HLA-restricted Mtb-reactive T cell metaclones defined by sequence motifs[15,28]. However, selecting an appropriate clustering method for a given dataset remains a challenge, with no broad consensus on best practices[29,30]. A key difficulty lies in evaluating clustering performance, particularly in balancing the trade-off between sensitivity (for clusters often measured by retention) and positive predictive value (for clusters typically assessed as purity). Effective benchmarking of clustering algorithms, therefore, requires comparing the maximum achievable purity at fixed levels of retention. However, many existing tools generate only a single clustering solution and lack flexibility to produce clusters at multiple resolutions.

To address this limitation, we developed Metaclonotypist, a modular pipeline for metaclone discovery that allows easy substitution of sequence similarity measures, thresholding choices, and clustering algorithms. Using published sets of TCR β sequences of known epitope specificity from the VDJdb database[23], we used Metaclonotypist to systematically identify Pareto optimal algorithmic approaches (Fig. 4A). Using connected component clustering, we found that adjacency graphs constructed by thresholding pairwise CDR3 edit distances were inferior to those built using TCRdist, a metric that incorporated both CDR3 and V-gene similarity (Fig. 4B). At high levels

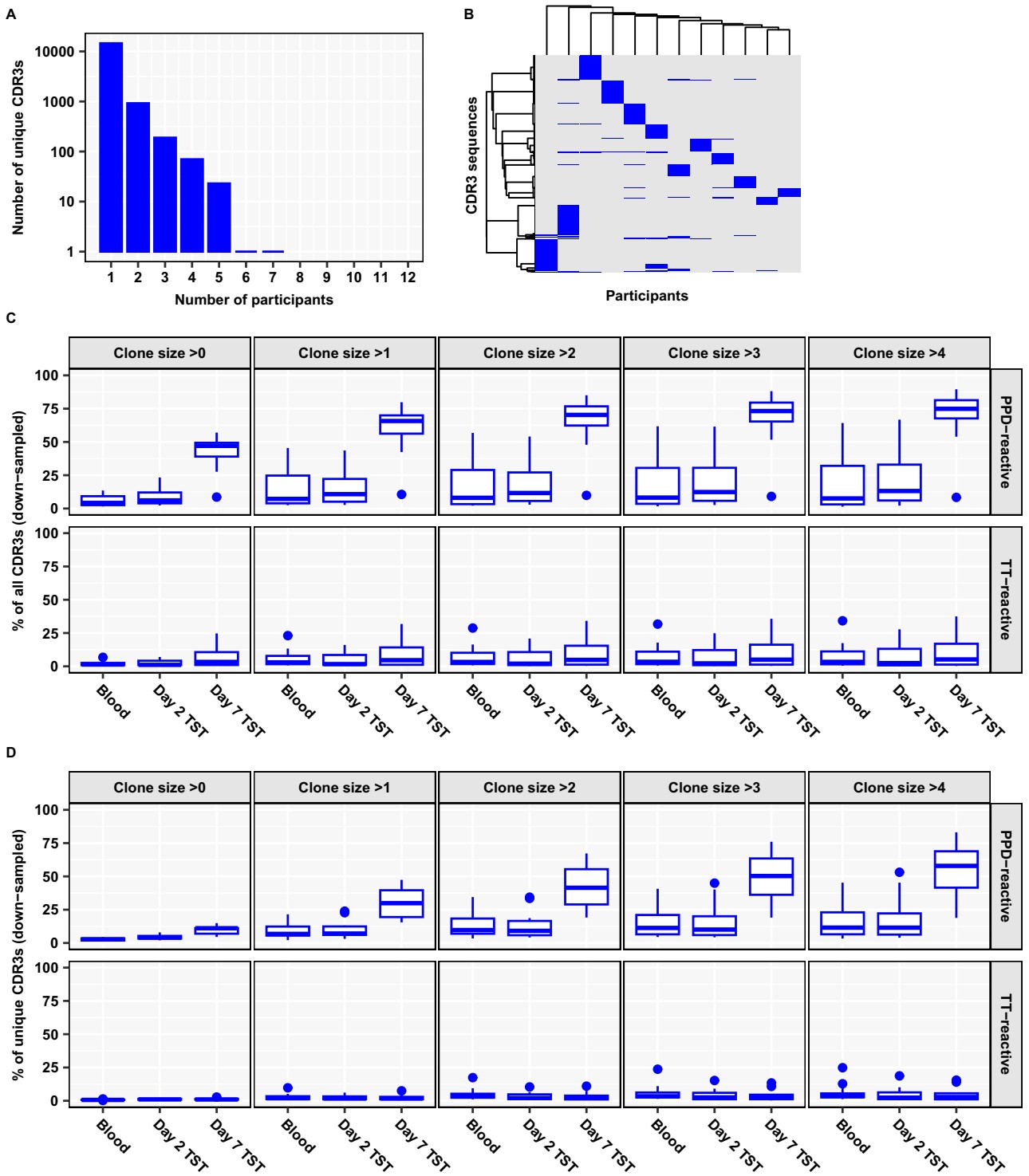

**Fig. 3 | Expansion of diverse PPD-reactive TCRs in Day 7 TST. A** Histogram of number of unique PPD-reactive β-chain CDR3s shared by different numbers of participants, in ex vivo PPD-stimulated PBMC from a subset of the study population (N = 12). **B** Heatmap of unique ex vivo PPD-reactive β-chain CDR3s (clustered by Ward D2 linkage). Each column across the x-axis represents one participant. Abundance of total (**C**) or unique (**D**) ex vivo PPD- or TT-reactive β-chain

CDR3 sequences in blood (n = 12), Day 2 TST (n = 11) and Day 7 TST (n = 10). Individual β-chain repertoires from blood and TSTs were down-sampled to 16,000 total TCRs and stratified by clone size ( = TCR count). The boxplots display median and inter-quartile range (IQR), with outlier data points (more than 1.5 × IQR beyond the box hinges) shown as dots.

of retention, we observed that Leiden clustering outperforms simple connected components clustering, as it breaks down large components into more modular, coherent clusters.

Based on these findings, we applied Leiden clustering to TCRdist adjacency graphs to analyse the day 7 TST TCR β repertoires that are

highly enriched for *Mtb*-reactive T cells, but for which the specific epitopes being recognised are unknown. To ensure scalability given the large number of TCRs to be clustered in our dataset, we leveraged our previously developed symmetric deletion lookup algorithm to rapidly identify candidate TCR neighbours in adjacency graphs[31].

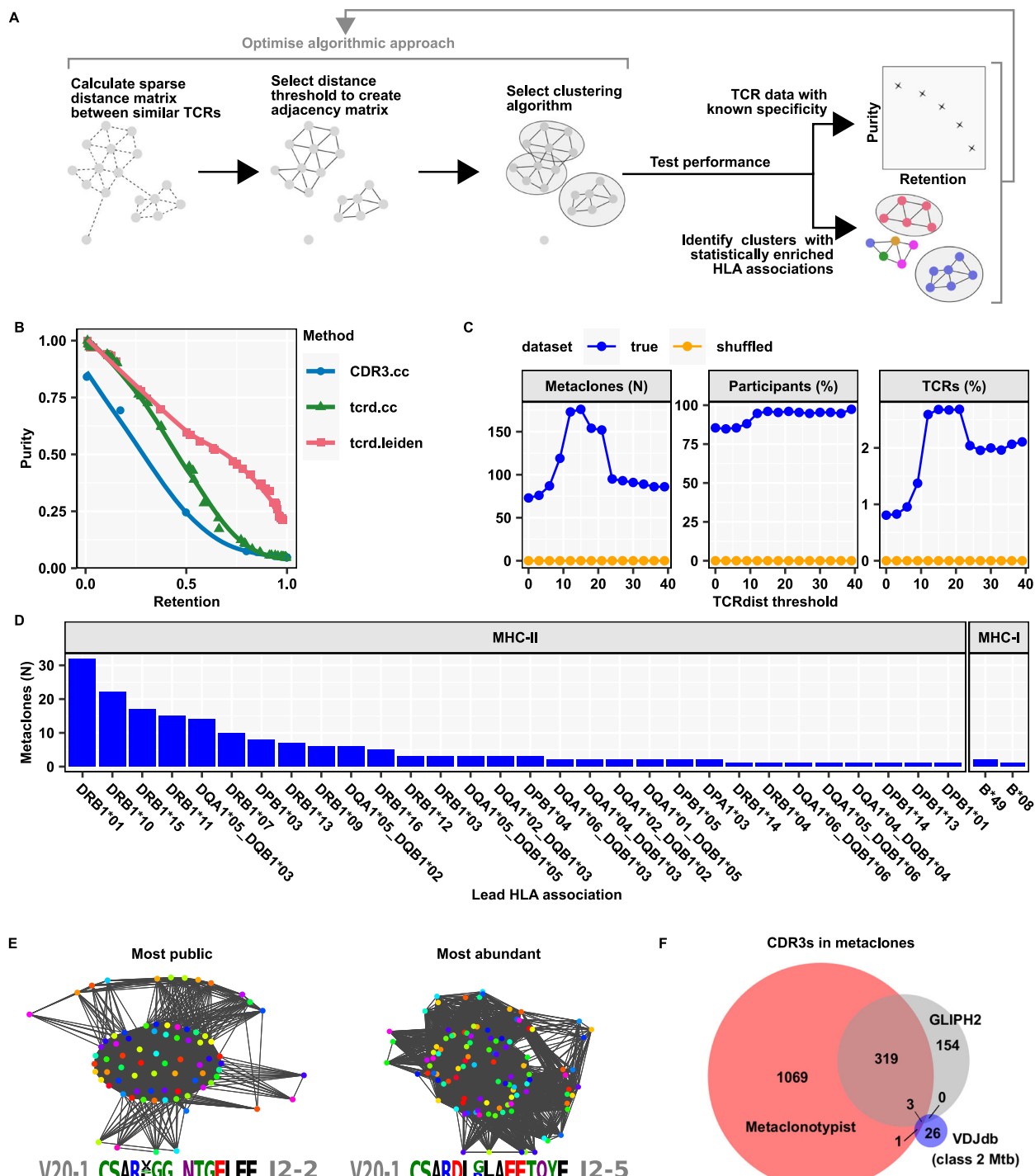

**Fig. 4 | Discovery of public HLA-restricted TCR metaclones from Day 7 TSTs. A** Schematic overview of the Metaclonotypist analysis pipeline and evaluation in data with known (purity/retention), or unknown specificities (by identifying significant enrichment of HLA associations, detailed in Supplementary Data 4). **B** Trade-off between cluster purity and retention for different clustering algorithms and threshold choices benchmarked on 4840 TCRs specific to 22 distinct pMHCs from VDJdb. Metaclonotypist using TCRdist scores and Leiden clustering provides Pareto optimal clustering. **C** Number of HLA-enriched metaclones (left hand plot), percentage of contributing participants (middle plot) and percentage of contributing unique TCRs (right hand plot) identified at varying TCRdist thresholds by Metaclonotypist analysis of day 7 TST TCR β-chain repertoires (*N* = 151, sub-sampled to between 5000 and 10,000 TCRs per repertoire) using true or shuffled HLA allele associations. **D** Frequency distribution of the most significant HLA associations for each β-chain Metaclonotypist metaclone, stratified by HLA class II (*n* = 177) and class I (*n* = 3) allele enrichment. **E** Exemplar adjacency graphs and TCR sequence motifs of most public (found in 82 out of 128 participants) and most abundant (matching 5197 out of 288,000 TCRs) day 7 TST Metaclonotypist meta-clones from down-sampled repertoires (*N* = 128 with 16,000 TCRs from each repertoire). Each node represents a single TCR stratified by distinct donors (colours). **F** Venn diagram showing the overlap of unique β chain CDR3 amino acid sequences included in class II-associated metaclone clusters by Metaclonotypist or GLIPH2 and annotated as class II-restricted *Mtb*-reactive TCRs in VDJdb.

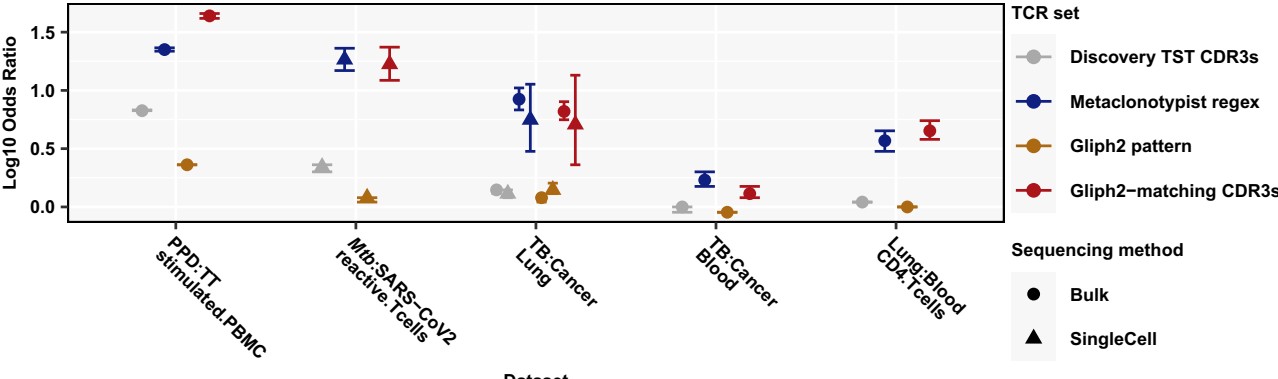

**Fig. 5 | Validation of Mtb reactivity and publicity of metaclones.** Relative enrichment of β-chain TCR sets derived from expanded (TCR count >1) day 7 TST repertoires (122,253 TCR clones, 151 individuals) in multiple external data sets showing odds ratio (OR) point estimates with 95% confidence intervals for the pairwise comparisons indicated. PBMC (bulk-TCRseq): in vitro stimulation of PBMC from $n = 12$ individuals with either purified protein derivative (PPD) from *Mtb* or tetanus toxoid (TT), followed by bulk TCR sequencing (see Fig. 3), comprising $n = 7,743,878$ PPD-stimulated and $n = 1,453,823$ TT-stimulated TCRs. T-cells (sc-TCRseq): in vitro stimulation of PBMC with either *Mtb* lysate ($n = 70$)[15] or SARS-CoV2 ($n = 16$)[52], followed by flow cytometric sorting and single cell TCR sequencing of activated T cells, resulting in $n = 21,212$ *Mtb*-reactive and $n = 149,208$ SARS-CoV2-reactive T cells. Lung (sc-TCRseq): lung tissue resections from TB patients ($n = 5$)[53]

or lung cancer patients ($n = 3$)[54], followed by single cell TCR sequencing, resulting in $n = 20,025$ TB-associated and $n = 17,019$ cancer-associated T cells. Lung (bulk-TCRseq): lung tissue resections from TB patients ($n = 13$) or lung cancer patients ($n = 3$), followed by bulk TCR sequencing, resulting in $n = 1,615,131$ TB-associated and $n = 218,372$ cancer-associated β TCRs. Blood (bulk-TCRseq): bulk TCR sequencing of whole blood samples from TB patients ($n = 11$) or lung cancer patients ($n = 4$), resulting in $n = 1,081,593$ TB-associated and $n = 735,834$ cancer-associated β TCRs. CD4-T (bulk-TCRseq): bulk TCR sequencing of CD4 T cells, flow-sorted from lung tissue or blood samples from TB patients ($n = 5$), resulting in $n = 336,787$ lung-derived and $n = 219,541$ blood-derived β TCRs (detailed in Supplementary Data 9)[55].

We selected the TCRdist threshold, which identified the largest number of TCR clusters in our dataset that were significantly enriched for individuals with a shared HLA allele (Fig. 4C). The stringency of the false discovery rate control was tested by showing that random shuffling of the HLAs associated with each individual returned no HLA-enriched T cell metaclones regardless of TCRdist threshold. For the TCRβ repertoires, the optimal TCRdist threshold was 15, identifying 180 HLA-associated metaclones, of which 177 were restricted by HLA class II molecules. This is consistent with the known predominance of CD4 T cells in the PPD-reactive TST response, which are restricted by HLA class II. Among these, HLA-DRB1 associated metaclones were most frequent (Fig. 4D).

At the optimal threshold, >95% of study participants contributed to at least one T cell metaclone (Fig. 4C, middle panel). The identified single chain metaclones can be represented as sequence motifs and visualised as adjacency graphs in which labelling of the individual TCR sequence by the donor reveals substantial publicity (Fig. 4E and Supplementary Fig. 9B). We therefore consider metaclones a measure of public reactivity to an unknown peptide:MHC complex. However, only approximately 2.7% of unique TCR sequences were incorporated into HLA-associated metaclones, consistent with the hypothesis that the majority of *Mtb* reactive TCRs are private (Fig. 4C, right panel). We benchmarked our metaclone discovery pipeline against clusters identified by the GLIPH2 algorithm. After exploring different GLIPH2 filters (Supplementary Fig. 10A), we retained clusters which passed the same stringent HLA association test with multiple testing correction applied in the Metaclonotypist pipeline. These GLIPH2 clusters (Supplementary Data 7, 8) also primarily associated with class II alleles, which demonstrates the robustness of our findings with respect to the algorithmic approach. However, Metaclonotypist clustered almost three times as many unique CDR3s compared to GLIPH2, suggesting that Metaclonotypist offers increased sensitivity to detect metaclone associations (Fig. 4F). Indeed, comparison of CDR3s clustered by Metaclonotypist or GLIPH2 with VDJdb MHC class II-restricted β-chain CDR3 entries annotated as *Mtb*-reactive found three matches in the GLIPH2 output, but an additional match in the Metaclonotypist output (Fig. 4F and Supplementary Fig. 9A).

## Validation of Mtb-reactivity of day 7 TST-derived T cell metaclones and population level immunodominance of Mtb-derived epitopes

To confirm that our day 7 TST-derived, class II associated T cell metaclones represented public *Mtb*-reactive T cell responses, we calculated their enrichment in independent TCR sequencing data derived from people with TB compared to other diseases, or at the site of TB disease compared to blood. Enrichment in this analysis suggests specificity for *Mtb*-reactive TCRs. For internal validation, we first confirmed that the day 7 TST metaclones identified by Metaclonotypist were significantly enriched in PBMC stimulated with PPD compared to PBMC stimulated with tetanus toxoid from the same study population (Fig. 5). For external validation, we then showed they were significantly enriched in peripheral blood single cell sequencing data derived from patients with TB compared to SARS-CoV-2 infection; bulk TCR sequencing of blood and lung tissue from patients with TB compared to cancer diagnoses; single cell TCR sequencing of lung tissue from patients with TB compared to cancer diagnoses; and in bulk TCR sequencing of CD4 T cells from the site of pulmonary TB disease compared to blood of the same patients (Fig. 5). We benchmarked this analysis for publicity and *Mtb*-reactivity of metaclones identified using Metaclonotypist against metaclones identified by the GLIPH2 algorithm. Metaclonotypist motifs showed comparable enrichment to CDR3β sequences from GLIPH2 clusters with statistically significant class II HLA-allele associations (Fig. 5; blue vs. red), suggesting that Metaclonotypist retains similarly high specificity, despite clustering a larger proportion of TCRs into metaclones (Fig. 4F). Additional GLIPH2 filters did not improve performance (Supplementary Fig. 10B). Importantly, amino acid metaclone motifs as defined by GLIPH2 were not substantially enriched in TB samples in these datasets (Fig. 5; light-brown), indicating the need for more restrictive motif definitions. Interestingly, compared to metaclones, we found substantially lower enrichment of the full discovery set of expanded day 7 TST β TCRs (Fig. 5; grey vs. blue), suggesting that a substantial proportion of day 7 TST TCR clones may not be specific to *Mtb*, and that identification of metaclones significantly improves antigen-agnostic enrichment of the *Mtb*-reactive T cell response.

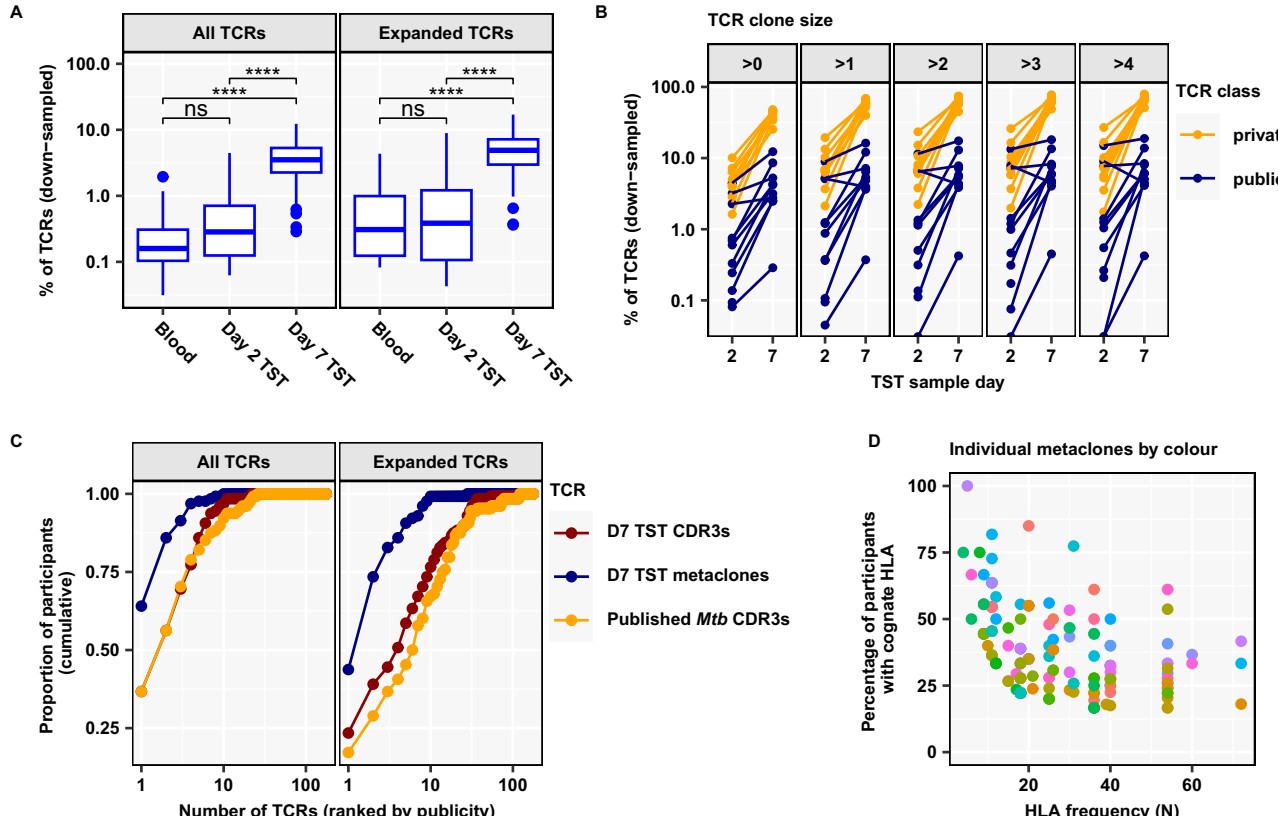

**Fig. 6 | Mtb-reactive metaclones constitute a small proportion of the Day 7 TST repertoire but capture the most public response. A** Abundance of HLA class II-restricted Metaclonotypist β-chain metaclones in TCR repertoires (each down-sampled to 16,000 TCRs; $n = 20$ Blood, $n = 16$ Day 2 TST, $n = 128$ Day 7 TST), shown as percentage of all TCRs or of all expanded ( >1) TCRs. Boxplots display median and inter-quartile range (IQR), with outlier data points (more than 1.5 × IQR beyond the box hinges). Statistical significance was assessed with unpaired, two-sided Wilcoxon tests and corrected for multiple testing (ns FDR > 0.05, * FDR < 0.05, ** FDR < 0.01, **** FDR < 0.0001). **B** Abundance of public and private Mtb-reactive CDR3s in the same individual quantified as percentage of all Day 2 or all Day 7 TST TCRs, stratified by clone size ( = TCR count). β−chain TCRs were classified as public if they matched a Metaclonotypist metaclone, or else as private if they matched a private PPD-reactive CDR3 sequence identified from ex vivo stimulated PBMC from the same individual (see Fig. 3). This analysis was restricted to individuals with paired in vitro stimulation experiments ($n = 11$ Day 2 TST, $n = 10$ Day 7 TST), and performed on repertoires down-sampled to 16,000 TCRs each. **C** HLA-class II restricted Metaclonotypist β-chain metaclones (blue), CDR3s with published Mtb reactivity (yellow), and CDR3s present in Day 7 TSTs (red) were each ranked by their publicity across 128 Day 7 TST β repertoires (each down-sampled to 16,000 TCRs) and plotted against the cumulative proportion of participants expressing the TCR. Presence of TCRs was assessed using either all TCR sequences in each sample or only expanded TCRs (present more than once). **D** Proportion of participants with a cognate HLA contributing to a metaclone in the discovery dataset ($n = 151$ Day 7 TST) and frequency of the cognate HLA for each HLA class II-restricted Metaclonotypist metaclones represented by individual points.

Having confirmed that class II associated day 7 TST Metaclonotypist metaclones represented public Mtb-reactive TCRs, we investigated their emergence in the TST over time. Despite some evidence for between-donor convergence of TCR sequences suggesting emergence of public T cell responses in the day 2 TST (Fig. 2D), enrichment of T cell metaclones at day 2 compared to blood did not reach statistical significance (Fig. 6A). In contrast, at day 7 metaclones were about tenfold enriched in the TST samples compared to blood and day 2 TST repertoires (Fig. 6A and Supplementary Fig. 11), and the percentage of day 7 TST repertoires captured by metaclones was positively correlated with the TST induration measured at day 2 (Supplementary Table 1). However, putative Mtb-reactive metaclones still only represented the minority of the total TCR sequence data at day 7. Consistent with this predominantly private response, we found no significant enrichment of previously published Mtb-reactive CDR3 sequences in the day 7 TST when these data were sub-sampled to the same number of CDR3s included within metaclones (Supplementary Fig. 12). Therefore, despite oligoclonal restriction of the T cell repertoire over time in the TST, the day 7 repertoire may retain a substantial breadth of epitope diversity. For the subset of the study population in which we had identified PPD-reactive CDR3s in ex vivo stimulated PBMC, we

compared the relative proportions of TST TCRs that either contained private PPD-reactive CDR3 sequences or clustered in public T cell metaclones at the level of individual participants. TCRs with private PPD-reactive CDR3 sequences were more frequent than public metaclone motifs in both day 2 and day 7 TSTs across the range of TCR clone sizes (Fig. 6B and Supplementary Fig. 13A). There was no statistically significant difference between expansion of private PPD-reactive and public metaclone TCRs between day 2 and day 7 TSTs, within the limitations of the sample size in this analysis (Supplementary Fig. 13B, C). Despite the dominance of private PPD-reactive TCRs at individual level, evaluation of the cumulative publicity of metaclones ranked by their publicity suggests that as few as 10 metaclones are sufficient to identify Mtb-T cell reactivity in the entire study population (Fig. 6C, Table 3, and Supplementary Fig. 14). At the level of individual metaclones, there was substantial heterogeneity in publicity even among individuals with the cognate HLA allele, offering the possibility of testing Mtb-reactive T cell metaclone responses as correlates of protection and pathogenesis (Fig. 6D).

Finally, we stratified participants into groups that were expected to be enriched for either recent or remote exposures, based on epidemiological criteria. We found a modest increase in TST induration in

**Table 3 | Top 10 public metaclones in down-sampled D7 TST dataset**

| Index | Publicity | Consensus CDR3aa | V gene usage | Lead HLA association (odds ratio, p-value) |
|---|---|---|---|---|
| 8 | 82/128 | CSARVGGNTGELFF | TRBV20-1 | DRB1*15 (10.2, 1.84E-09) |
| 76 | 78/128 | CSAGGLAGNEQFF | TRBV20-1 | DQA1*01_DQB1*05 (8.6, 9.30E-06) |
| 13 | 69/128 | CASSLGSVSYEQYF | TRBV7-9 | DRB1*15 (11.3, 3.55E-09) |
| 21 | 64/128 | CSARDLGLAEETQYF | TRBV20-1 | DRB1*04 (11.3, 4.41E-08) |
| 39 | 60/128 | CSVGETQYF | TRBV29-1 | DQA1*05_DQB1*03 (9.1, 4.63E-07) |
| 33 | 56/128 | CSARAGYGYTF | TRBV20-1 | DRB1*10 (44.0, 2.61E-07) |
| 91 | 50/128 | CASSLEGETQYF | TRBV7-9 | DRB1*11 (6.2, 2.17E-05) |
| 6 | 49/128 | CASSRGAQTYEQYF | TRBV18 | DPB1*04 (27.5, 8.91E-10) |
| 100 | 47/128 | CSARGQGNEQFF | TRBV20-1 | DRB1*13 (7.9, 2.73E-05) |
| 97 | 43/128 | CASSPGRETQYF | TRBV6-6\| TRBV6-5\| TRBV6-9 | DRB1*10 (26.8, 2.39E-05) |

the recent exposure group, but found no differences for any of the TST metrics measured on the molecular level (Supplementary Fig. 15).

## Discussion

Our study reports the temporal evolution of the TST response at the molecular level. We apply this model to establish the in vivo clonal repertoire of *Mtb*-reactive T cell responses. At the level of individual TCRs, we found this response to be almost entirely idiotypic, with very little inter-individual sharing of TCR chains even between individuals with shared HLA. Therefore, we introduce an analytical pipeline we have called Metaclonotypist for identification of public T cell metaclones: clusters of distinct TCRs across different individuals predicted to share peptide-MHC specificity. They enable antigen-agnostic identification of T cell responses to immunodominant targets at population level by overcoming inter-individual TCR sequence diversity. We find predominantly non-antigen-specific T cell recruitment to the TST initially, co-incident with peak inflammatory responses. These are then replaced by predominantly *Mtb*-reactive T cells through selected oligoclonal T cell proliferation. T cell metaclones derived from the day 7 TST reveal public T cell responses that are highly enriched in multiple sources of TCR sequence data from the blood and lung tissue of patients with TB compared to other diseases, and at the site of pulmonary TB compared to blood. The enrichment of TST metaclones in the TB lung suggests that T cells recruited into PPD-challenged skin have clinical relevance and supports the use of the TST as experimental human challenge model for immune responses in TB. The vast majority of TCRs enriched in the day 7 TST are not incorporated into public metaclones but remain private to individuals. Nonetheless, the cumulative publicity of the most public metaclones indicates striking population-level coverage of *Mtb* reactivity. Although metaclones are defined by their β-chain motifs only, we hypothesize that this reflects immunodominance of specific *Mtb* epitopes. This finding extends previous studies that report generalisable antigenic immunodominance at the protein level[12–14] by providing a scalable antigen-agnostic approach to potentially identifying generalisable immunodominance at the level of specific peptides.

The rapid accumulation of T cells at the site of the day 2 TST, which occurs at the time of maximum clinical inflammation but before evidence of cell proliferation, indicates recruitment of cells from the circulating pool. Recruitment appears to be non-selective, demonstrated by the similar proportions of *Mtb*-, tetanus toxoid-, CMV- and EBV-reactive CDR3 sequences observed in the day 2 TSTs and blood samples. However, *Mtb*-specific sequences are enriched relative to CMV-, EBV- and tetanus toxoid-reactive CDR3 sequences in day 7 TSTs, suggesting that the proliferation response is not simply due to bystander proliferation of all local T cells. Our data cannot fully discriminate the extent to which reduction of non-*Mtb*-reactive T cells at the TST site by day 7 is due to their clearance or their relative decrease caused by the expansion of *Mtb*-reactive TCRs dominating the sampled repertoire. The latter process is supported by our observation that the full set of day 7 TST TCRs shows substantially lower enrichment in independent TB-associated datasets compared to day 7 TST metaclones. We hypothesise that local antigen causes *Mtb*-reactive cells to be retained at the TST site and then proliferate. Importantly, our approach therefore overcomes a limitation of peripheral blood, by sampling the TCR repertoire following recruitment from the whole in vivo T cell repertoire. Inflammatory induration in the TST is known to be dependent on prior expansion of *Mtb* T cell memory. Yet, we found no enrichment of previously reported *Mtb*-reactive sequences, public *Mtb*-reactive metaclones or private PPD-reactive CDR3s alongside peak inflammatory responses in day 2 TSTs compared to blood. Interestingly, however, within individual donors, there was a significant increase in CDR3 sequence convergence at the amino acid level in day 2 TST samples compared to blood. This finding suggests some degree of *Mtb*-specific T cell selection driving the TST response, albeit below the limit of sensitivity to detect through enrichment compared to blood. We also observed a positive correlation between TST induration on day 2 and the extent of predicted *Mtb* reactivity on day 7 (measured as proportion of metaclones, or proportion of expanded TCRs), consistent with the hypothesis that both arise from a common determinant of the TST response, likely to be the frequency of circulating *Mtb*-reactive memory T cells that initially drive inflammatory induration and then proliferate in response to antigen. Substantially less CDR3 convergence was evident between donors. This was partly explained by inter-individual diversity of MHC alleles but was also driven by the enormous potential diversity of the TCR repertoire. Interestingly, we found a much stronger relationship of inter-individual CDR3 convergence to increased sharing of HLA class II alleles than to increased sharing of HLA class I alleles. These data suggested that CD4 T cells are driving the maturation of T cell repertoire in the TST. This interpretation is consistent with the finding that the transcriptional signature for CD4 T cells, but not CD8 T cells, increases between day 2 and day 7 TST and correlates best with the transcriptional signature for cellular proliferation. Likewise, it is consistent with the observation that day 7 TST metaclones were almost exclusively restricted to HLA-class II alleles. Hence, we conclude that the TST is predominantly a model for CD4 T cells responses, as previously reported for T cell responses to *Mtb* stimulation of peripheral blood cells[15]. However, we note that this does not exclude a role for CD8 T cell responses in TB. In addition to evidence from other models that CD8 T cells may contribute to protective immunity, bulk transcriptional profiling of the TST response presented here and our recent report of single cell sequencing analysis of the TST response[32] reveal substantial enrichment of activated CD8 T cells in the day 2 TST. The lack of further enrichment of CD8 T cell responses in the day 7 TST may be a limitation of the TST model. Nonetheless, the advantage of the skew towards CD4 T cell responses is highlighted by recent evidence that CD4 T cell responses make a more important contribution than CD8 T cells to vaccine-mediated immunity[33].

Inter-individual variation in total T cell reactivity and/or the clonal repertoire, including intra- and inter-donor co-incidence metrics in the TST, may relate to duration of *Mtb* infection or time interval since immunological clearance of bacteria. While neither of these variables

can be measured accurately, epidemiological classification of our study cohort into household contacts, likely to be enriched for recent exposures, and occupational health screening participants, likely to be enriched for remote exposures, showed no differences in any of the measured molecular TST metrics.

By developing a modular computational pipeline for metaclone discovery, we were able to systematically identify algorithmic approaches with superior positive predictive value through controlled comparisons at fixed sensitivity. In combination with previously developed fast sequence similarity search tools[31], Metaclonotypist scales accurate TCRdist-based clustering to large datasets and allows flexible optimisation of parameters. The optimised Metaclonotypist-pipeline identified significantly more HLA-associated metaclones in our dataset than the principal metaclone discovery algorithm (GLIPH2) that has been previously used to identify *Mtb*-reactive metaclones[15,28], while maintaining comparable enrichment in a wide-ranging external validation. Despite this increased sensitivity, the majority of day 7 TST TCRs remained excluded from Metaclonotypist metaclones, which we have labelled as private T cell responses. We anticipate that further artificial-intelligence driven improvements in measures of TCR sequence similarity[34] or corrections for recombination biases[27] will allow clustering of more distant TCRs with shared peptide-MHC specificity. Therefore, our current approach provides only a minimum estimate of publicity in the *Mtb*-reactive T cell repertoire. Even so, as few as 10 metaclones were sufficient to identify generalisable *Mtb* reactive T cell responses in our study population of ≥128 individuals. Interestingly, for most metaclones, not all individuals with the restricting HLA allele had detectable metaclone TCRs. This finding is compatible with potential inter-individual heterogeneity of *Mtb* responses to particular pMHCs, which in conjunction with future improvements in metaclone discovery and deeper TCR sequencing, may allow linking T cell metaclone responses to differential outcomes of *Mtb* infection.

Consistent with most data on HLA class II restriction of CD4 T cell responses, we found the lead HLA associations for day 7 TST meta-clones to be predominantly HLA-DR alleles. This skew is attributed to the higher prevalence of diverse DR alleles, a structure that allows them to bind a wider variety of peptides, and potentially higher levels of expression than HLA-DP and HLA-DQ alleles on antigen presenting cells[35–37]. Several metaclones were significantly associated with more than one HLA allele, likely due to the co-expression of HLA alleles in linkage disequilibrium[38], limiting the ability to resolve which allele is the restricting MHC for such metaclones. The application of bulk TCR sequencing maximised the depth of data and therefore the sensitivity of our analysis. Our primary approach focussed on β-chain sequences to leverage their greater diversity and provide greater discrimination of the TCR repertoire than possible by analysis of α chains. Reassuringly, analyses of α-chain sequences replicated the findings from β-chain repertoires. In future work, identification of public metaclone motifs in single cell sequencing datasets can identify associated αβ-chain pairs. Although the more limited depth of single cell sequencing limits sensitivity for metaclone discovery, αβ-chain pairs are necessary to pursue reverse epitope discovery to test the prediction that meta-clone clustered TCRs share epitope specificity, develop functional T cell assays to evaluate the association of immunodominant T cell metaclones with clinical outcomes of infection, and enable innovation in vaccine design based on specific protective epitopes in place of protein antigens.

Our application of Metaclonotypist, combining fast clustering and HLA associations, to TCR sequence data from the day 7 TST that is highly enriched for *Mtb*-reactive T cells recruited from the in vivo repertoire, has identified a catalogue of metaclones, which together span the majority of the study population. We hypothesise that these metaclones will provide powerful approaches to improve disease-risk stratification in *Mtb*-infected people, and, in combination with single-

cell sequencing, will enable identification of epitopes to resolve pro-tective and pathogenic T cell immunity critical to the development of more effective vaccines.

## Methods

### Study approvals
Research ethics and regulatory approvals for the present study were provided by UK National Research Ethics Service (NRES) Committee (Fulham) reference nos 11/LO/1863 and 18/LO/0680, and the NRES Committee (Camden and Islington) reference no 14/LO/0505. All study participants provided written informed consent. Participants were compensated for travel expenses.

### Study population and sampling
Study participants comprised healthy HIV seronegative adults, 18–60 years of age, with immune memory for *Mtb*-specific antigens identified by positive peripheral blood IFNγ release assays using the Quanti-FERON Gold Plus Test, but no clinical or radiological evidence of active tuberculosis. Male and female participants were included in this study, and sex was determined by self-reporting. The analyses were not dis-aggregated by sex because this was not a component of the research question. Peripheral blood mononuclear samples purified by Ficoll-Paque Plus (GE Healthcare Biosciences) density gradient centrifuga-tion of whole blood, were collected on participant enrolment, and cryopreserved in foetal calf serum (FCS, Sigma) supplemented with 10% DMSO (Sigma). Blood samples were collected into Tempus RNA preservative tubes (Thermo Fisher Scientific, cat no. 4342792) and total RNA extracted with the Tempus Spin RNA Isolation Kit (Ambion; Life Technologies, cat no. 4380204). Globin mRNA and genomic DNA were removed using the GlobinClear kit (Thermo Fisher Scientific, cat no. AM1980) and TURBO DNAfree kit (Ambion, Life Technologies, cat no, AM2238). Participants then received 0.1 ml intradermal injections of either 2U tuberculin (Serum Statens Institute) into the volar aspect of each forearm, or control saline in one arm[9,17,19]. The precise injection sites were marked with permanent marker pen and used to position the site of 3 mm punch biopsies at designated time points. Saline controls were biopsied on day 2, TST sites were biopsied on day 2 at one site and on day 7 at the contralateral site. Skin biopsies were placed in RNALater (Qiagen) and cryopreserved at −70 °C, prior to RNA extraction. Skin biopsies were thawed at room temperature for 30 min and transferred into CK14 lysing tubes (Bertin Instruments) containing 350 μl Buffer RLT supplemented with 1% 2-mercaptoethanol (Sigma). Samples were homogenized on a Precellys Evolution instrument for 6 cycles of 23 s at 6300 rpm, with 2 min cooling on ice between cycles. Following centrifugation to pellet debris and beads, total RNA was purified from the supernatant using the RNeasy Micro Kit (Qiagen, cat no. 74004), according to the manufacturer's instructions. Con-taminating DNA was removed using the TURBO DNAfree kit (Ambion, Life Technologies, cat no. AM2238). In a sensitivity analysis, partici-pants were stratified by the indication for QuantiFERON testing, as surrogate for possible interval time from exposure. In this context, household contacts of active TB index cases were expected to be enriched for individuals with more recent exposures than individuals being tested via routine occupational health screening.

Participant demographics were collected in a RedCAP database, are summarised in Tables 1 and 2, and detailed on a per-sample level in Supplementary Data 1.

### RNA sequencing and analysis
Total RNA from TSTs was subjected to genome wide mRNA sequen-cing. cDNA libraries were generated using the KAPA Hyperprep kit (Roche, cat no. 07962363001), and sequencing was performed on the Illumina Nextseq using the Nextseq 500 High Output 75 cycle kit (Illumina, cat no. 20024906) according to manufacturers' instructions, providing a median of 22 million (range 10–50 million) 41 bp paired-

end reads per sample. RNAseq data were mapped to the reference transcriptome (Ensembl Human GRCh38 release 111) using Kallisto (v0.46)[39]. Transcript-level counts were summed on gene level, and annotated with Ensembl gene ID, gene symbol and gene biotype using the R/Bioconductor packages tximport and BioMart. Raw counts of 23,820 Ensembl gene IDs, retained after exclusion of pseudogenes, were used for differential expression analysis with the SARtools (v1.8.1) implementation of DeSeq2 (v1.42.1)[40], with a false discovery rate (FDR) < 0.05 and log2 fold difference of ≥1. For all other analyses, raw counts were converted into transcripts per millions (TPM) values, and log2 transformed after the addition of a pseudocount of 0.001. Duplicated gene symbols were filtered by retaining the gene with highest expression per sample.

Upstream regulator analysis of the differentially expressed genes was performed using Ingenuity Pathway Analysis (Qiagen). This was visualised as a network diagram using the Force Atlas 2 algorithm in Gephi v0.9.4, and used to derive co-regulated gene-expression networks[41]. This analysis was restricted to upstream regulators predicted to be significantly activated (Z-score > 2, adjusted $p$-value < 0.05), targeting at least 4 downstream genes, and annotated with one of the following functions: cytokine, kinase, transmembrane receptor, and transcriptional regulator, representing the canonical components of pathways which execute transcriptional reprogramming in immune responses. For each upstream regulator, pairwise Spearman correlations of the TPM expression values of the target genes were calculated among TST samples. Upstream regulators were selected as significant if the average co-correlation was significantly (FDR < 0.05) greater than the distribution of average correlation coefficients obtained from 100 iterations of selecting an equivalent number of random genes. Reactome pathway enrichment of differentially expressed genes was analysed with the XGR (v1.1.9) R package[42]. For visualisation, 20 pathway groups were identified by hierarchical clustering of Jaccard indices to quantify similarity between the gene compositions of each pathway. For each group, the pathway with the largest total number of genes was then selected to provide a representative annotation.

Transcriptional modules for T cell proliferation[41] and cell types present in the TST[32] have been derived and published previously. Their gene composition is listed in Supplementary Data 2. The expression of each module was quantified as the arithmetic mean log2 TPM value of its constituent genes.

## Ex vivo PBMC stimulation

Frozen PBMC were thawed at 37 °C, washed in RPMI-1640 media (Thermo Fisher Scientific) with 10% FCS, and resuspended at $10^6$ cells/ml in RPMI with 5% heat-inactivated male human AB serum (Sigma). $2 \times 10^5$ PBMC were seeded into individual wells of round bottom 96-well plates (Thermo Fisher Scientific) with one of 10 µg/mL purified protein derivative of *Mtb* (PPD; Serum Statens Institute), 100 µg/mL tetanus toxoid (TT; NIBSC), or control buffer for 6 days at 37 °C and 5% $CO_2$. At the end of this incubation period, plates were centrifuged ($400 \times g$ for 5 min) and the resulting cell pellets were lysed in RLT buffer (Qiagen). Samples from triplicate wells were pooled for RNA extraction using the RNeasy Micro kit (Qiagen, cat no. 74004). Up to 5 separate pooled samples were collected for each individual, for each stimulus.

## T cell receptor (TCR) sequencing and analysis

RNA extracted from skin samples, ex vivo stimulated PBMC, and peripheral blood Tempus tubes were subjected to sequencing of TCR α- and β-genes using an established quantitative TCR sequencing pipeline that integrates experimental library preparation and computational analysis with Decombinator V4[26,43,44], which defines and quantifies a TCR clone by its nucleotide sequence and reports its V, J and CDR3 annotation. To account for different sequencing depth between samples, repertoire metrics were calculated after downsampling all

samples to 16,000 unique molecular identified (UMI) reads (Supplementary Fig. 1). Since the UMI-based method quantifies the number of mRNA transcripts, we note that clone size does not measure the number of T cells directly. Nevertheless, mRNA number is a good proxy for clone size and clonal expansion since T cells do not substantially change TCR mRNA levels upon activation[26].

TCR α and β CDR3s from whole blood or skin biopsies were annotated as CMV, EBV or *Mtb*-reactive, if they were listed as sequences known to target these pathogens in the VDJdb TCR repository (https://vdjdb.cdr3.net/; accessed 01/10/2024)[23,45], the McPAS database (http://friedmanlab.weizmann.ac.il/McPAS-TCR/; accessed 16/09/2023)[22] or in Table S2 from Musvosvi et al.[15]. The collated antigen-reactive CDR3 sequences are summarised in Supplementary Data 3. To identify antigen-reactive CDR3s from in vitro cultures, we identified CDR3 sequences with ≥eightfold increased abundance in antigen stimulated, but not unstimulated PBMC compared to whole blood from the same individual[26]. CDR3s absent in matching blood were set to the median blood CDR3 abundance of 1 to allow expansion calculations for all in vitro CDR3s. To complement this analysis, we also used a more stringent definition of expanded T cell clones[41,46]. In this approach, CDR3s in Day 7 TST samples were defined as significantly expanded if their observed abundance was greater than expected using a Poisson distribution derived from Day 2 TST counts with FDR < 0.1%.

MAIT TCR enrichment was assessed based on their TCR α gene usage as sequences containing TRAV1-2, paired with TRAJ12, TRAJ20 or TRAJ33; iNKT TCRs were identified as TCRs containing TRAV10 paired with TRAJ18; and GEM TCRs were identified as TCRs containing TRAV1-2 paired with TRAJ9[47].

TCR repertoire diversity was assessed by the number of expanded TCR sequences (count>1), Gini index (repertoire inequality), and Hill Diversity indices. These diversity indices are defined as $D_q = \left( \sum_{i=1}^{R} p_i^q \right)^{\frac{1}{1-q}}$, where R is the number of distinct TCRs, $p_i$ the clonal frequency of the i-th clone, and q a parameter that determines the relative weight put on clonal abundance. We compared richness (total number of distinct TCRs), $D_0 = R$, Shannon Diversity (exponential of Shannon entropy), $D_1 = \exp\left(-\sum_{i=1}^{R} p_i \ln p_i\right)$, and Simpson diversity (inverse of Simpson's index), $D_2 = \frac{1}{\sum_{i=1}^{R} p_i^2}$. Among these measures Simpson diversity is most sensitive to clonal dominance, while Richness completely disregards variability in clonal expansions.

Within- and cross-donor convergence of TCR sequences was calculated as previously described[24]. Within-donor convergence was calculated as the proportion of all pairs of distinct clonotypes (as defined by nucleotide sequence identity) which were functionally convergent, i.e., that encode the same protein. We define $n_i$ as the number of distinct clonotypes encoding the $i - th$ TCR with i = 1, …, S, where $S$ is the number of unique clonotypes. We also define N = $\sum$ $n_i$ as the total number of clonotypes in the sample. We then can estimate the probability of coincidence within a sample as: $\hat{p_C} = \sum_{i=1}^{S} \frac{(n_i(n_i-1))}{N(N-1)}$. In comparing across samples, we define clonotypes by nucleotide sequence and donor identity. Defining $n_{I,1}$ and $n_{I,2}$ as the sampled counts of the $i - th$ TCR in donor 1 and donor 2, respectively, we estimate the probability of cross-donor convergence using: $\hat{p_C} = \sum_{i=1}^{S} \frac{n_{i,1} n_{j,2}}{N_1 N_2}$, where $N_1 = \sum_i n_{i,1}$ and $N_2 = \sum n_{j,2}$ is the total number of clonotypes in the two samples.

## HLA genotype imputations

DNA from participants was extracted from cryopreserved whole blood using the QIAamp spin column (Qiagen). Genotyping was conducted using the Illumina Infinium Global Diversity Array. HLA imputation was performed on the Michigan Imputation Server[48] using genotyped autosomal variants across the study population, filtered to include only SNPs with a minor allele frequency of ≥5% and a call rate of ≥95%.

Briefly, typed SNPs within the MHC region (6:27970031-33965553; hg19) were phased with Eagle (v2.4) and imputed using Minimac4 with the four-digit multi-ancestry HLA imputation reference panel (v2). Imputed SNPs with an imputation score ($R^2$) <0.8 were excluded, resulting in high-confidence HLA alleles for 158 individuals (Supplementary Data 4).

## Benchmarking TCR clustering approaches using Metaclonotypist

To compare TCR clustering approaches, we implemented Metaclonotypist, a modular computational pipeline for metaclonotype discovery, and benchmarked clustering performance using TCRs with known pMHC specificity from the VDJdb database[23] (Fig. 4A). Metaclonotypist proceeds in a series of steps (Fig. 4A). Metaclonotypist first calculates pairwise distances between TCRs according to sequence similarity metrics, from simple Levenshtein edit distances applied to the CDR3 sequence to more advanced metrics such as TCRdist. This first step can be optionally sped up by pre-filtering of candidate sequence neighbour pairs using the symmetric deletion lookup algorithm. It next generates an adjacency graph between sequences, by thresholding the pairwise sequence similarity with respect to a tuneable threshold. Each node in this graph represents a TCR found in an individual's repertoire, and edges connect all nodes with a similarity below the threshold. Within the graph Metaclonotypist then identifies putative metaclones by clustering. Clustering is performed using community detection algorithms as implemented in igraph[49]. Importantly, each step is modular and supports multiple choices to allow benchmarking of alternative approaches using different sequence similarity metrics, threshold choices for adjacency graph construction, and clustering algorithms.

To construct a benchmarking task, we selected data from all pMHCs with at least 220 associated TCR β sequences from VDJdb following filtering and data standardisation using tidytcells[50]. We then randomly down-sampled TCR repertoires from pMHCs with a greater number of sequences to obtain a dataset of 4840 TCR β sequences equally balanced across 22 pMHCs.

Clustering involves a multi-objective optimisation, with ideal clustering having both high purity and retention. To allow controlled comparisons across similarity metrics and clustering algorithms, we systematically varied distance thresholds for each method to be able to identify Pareto optimal solutions. We defined cluster purity as the weighted average of the dominant class frequency in each cluster: $\text{Purity} = \frac{1}{N}\sum_{k=1}^{K}\max_j |C_k \cap L_j|$, where $N$ is the total number of TCRs, $K$ the total number of clusters, $C_k$ the set of TCRs associated with cluster $k$, and $L_j$ the set of TCRs associated with label $j$ (here representing a specific epitope). We defined clustering retention as the fraction of all TCRs assigned to non-singleton clusters: $\text{Retention} = \frac{1}{N}\sum_{k=1}^{K} I(|C_k|>1) \cdot |C_k|$, where $I(|C_k|>1)$ is an indicator function that is one if $|C_k|>1$ and 0 otherwise.

Using this benchmarking approach, we compared connected component clustering of adjacency graphs based on CDR3 Levenshtein distance, which simply groups all TCRs connected by at least one edge into a cluster, to more advanced algorithms. Our results suggest that the more advanced TCR sequence similarity metric TCRdist is superior to simple Levenshtein distance calculated on the CDR3 alone. We furthermore found that Leiden clustering, which breaks up large connected components into multiple clusters where this increases cluster modularity, maintains higher purity at larger thresholds.

## Discovery of HLA-associated metaclones with Metaclonotypist

We considered metaclones as a set of β-chain TCR clones with an imputed common HLA-peptide specificity[27]. We identify putative metaclones by clustering TCRs based on sequence similarity and testing the HLA association of TCR clusters. We combined bulk-sequenced Day 7 TST repertoires for β chains. To reduce uneven sampling, we down-sampled large repertoires to 10,000 total counts and excluded repertoires with <5000 total counts. To increase confidence of restricting metaclone discovery to *Mtb*-reactive TCRs, only those with count >1 in these down-sampled repertoires were selected for analysis. TCRs with CDR3 amino acid length ≤5 were excluded from analysis. We then identified all pairs of TCRs that differ by ≤2 edits in their CDR3 hypervariable region using the symmetric deletion lookup algorithm[31]. We next calculated TCRdist scores between these pre-pruned TCR pairs using TCRdist3[27]. Based on our preliminary benchmarking, we used Leiden clustering for our identification of metaclonotypes in the day 7 TST (with parameters: resolution = 0.1, objective_function = "CPM", n_iterations = 4). After examining the effect of varying thresholds, we represented the TCR repertoire as an undirected graph based on the sparse adjacency matrix obtained by thresholding TCRdist scores ≤15.

Each cluster was tested for HLA association, by comparing the expression of specific HLA alleles by Fisher's exact test between two groups of individuals: those contributing at least one TCR to a cluster and the remainder of the population. Associations for HLA class II alleles (DP, DQ, DR) and class I alleles (A, B, C) were tested separately. HLA-association of metaclones with the DQ locus was tested with respect to all potential DQ heterodimers, by combining DQ alleles for the α and β HLA chain to account for the highly polymorphic nature of both the α and β chain of HLA DQ. P values were corrected for multiple testing using the Bonferroni-Hochberg procedure at a False Discovery Rate (FDR) of 0.1, where the number of tests was set equal to the product of the number of tested clusters and times the number of tested HLA alleles. To limit multiple testing, we only assessed association of clusters containing TCRs from ≥4 individuals with HLAs found in ≥4 individuals across the population. Where more than one significant HLA association was found, the most significant one was considered as lead HLA association. As a control, the link between HLA haplotype and individuals was randomly shuffled.

## Metaclone visualisation

Sequence logos were constructed in Python, using the seqlogos_vj plotting submodule of the pyrepseq package[51]. Graphs of TCR sequence similarity within a metaclone were visualised using Python bindings to the igraph package. Each node represents a TCR, and distinct colours are used to indicate donor origin. Nodes are connected by unweighted edges whenever corresponding TCRs were below the threshold of sequence similarity used for metaclone discovery.

## GLIPH2 analysis

GLIPH2 analysis was undertaken on the same set of Day 7 TST β-chain repertoires as described for metaclonotype discovery above, using default settings and CD48_v2.0 reference. The Metaclonotypist approach was then mirrored by selecting only GLIPH2 similarity clusters containing TCRs from ≥4 individuals and testing for associations with HLAs found in ≥4 individuals (done separately for class I and class II alleles). We explored the effect of filtering the GLIPH2 output further, as described before[15]. Filter 1 selected clusters that consisted of ≥3 unique CDR3s and had a Fisher_score, vb_score and length_score ≤0.05 each. Filter 2 applied the Fisher's exact test for HLA association, either with a significance threshold of $p < 0.05$ (as used by Musvosvi et al.[15]) or with an FDR < 0.1 as applied in the Metaclonotypist pipeline.

## Quantification of Metaclonotypist and GLIPH2 metaclones

Supplementary Data 5 and 6 list the HLA class II and class I restricted β metaclones, respectively, as identified by Metaclonotypist from day 7 TST samples. The tables include significant HLA allele associations for each metaclone, as well as the associated V gene(s) and a regular expression for the clustered CDR3 amino acid sequences. To identify

and quantify β-chain TCR sequences from various datasets that match a pre-defined class II associated metaclone in the context of the correct TCR chain, the V gene and CDR3 of each TCR was compared against the V gene and CDR3 regular expression of each metaclone.

Supplementary Data 7 and 8 list the HLA class II and class I restricted β chain GLIPH2 clusters, respectively, identified from day 7 TST samples. The tables include significant HLA allele associations, as well as the clustered CDR3 sequences and a regular expression for the shared CDR3 pattern. β-chain TCR sequences from various datasets were identified as matching a pre-defined class II associated GLIPH2 cluster in two different ways: (a) if their CDR3 amino sequence contained the regular expression of the GLIPH2 motif (GLIPH2 pattern G.T was excluded from analysis to increase specificity), (b) if their CDR3 amino acid sequence was part of the GLIPH2-clustered set of CDR3 sequences.

### Datasets for external validation of TST-derived metaclones

Processed single cell TCR sequencing data from activated T cells following in vitro stimulation of PBMC from $n = 70$ individuals with *Mtb* lysate were accessed from Supplementary Table S2 in the publication by Musvosvi et al.[15] Only good quality cells (flag = GOOD) were included, resulting in 21,212 cells with β-chain data.

Processed single cell TCR sequencing data from activated T cells following in vitro stimulation of PBMC from $n = 16$ individuals with SARS-CoV2 were provided by Lindeboom et al[52]. All longitudinal samples per patient were included, resulting in 149,208 cells with β-chain data.

Single cell TCR sequencing FASTQ data from human lung of $n = 5$ TB patients[53] were downloaded from the NCBI Gene Expression Omnibus resource (GSE253828) and processed with 10x Genomics CellRanger (v7.1.0) using the vdj pipeline and VDJ-T reference version 7.1. Single cell TCR data from filtered_contig_annotations.csv output files were integrated across all patients, resulting in 20,025 cells with β-chain data.

Single cell TCR sequencing FASTQ data from human lung of $n = 3$ lung cancer patients[54] were downloaded from the NCBI Gene Expression Omnibus resource (GSE154826) and processed with 10× Genomics CellRanger (v7.1.0) using the vdj pipeline and VDJ-T reference version 7.1. Single cell TCR data from filtered_contig_annotations.csv output files were integrated across all samples (including tumour and normal lung tissue) from all patients, resulting in 17,019 cells with β-chain data.

Processed bulk TCR sequencing data for lung tissue and whole blood from TB patients and cancer controls, as well as for sorted CD4 T cells from TB lung and TB blood were provided on Adaptive Biotechnologies' ImmunoSEQ website (https://clients.adaptivebiotech.com/). The cohort has been previously described[55], and an overview of utilised files and their metadata is provided in Supplementary Data 9. Only functional TCR sequences were included (sequenceStatus = In), and the vMaxResolved column was used as V gene annotation, but with the allele information excluded (e.g., TCRBV06-01*01 became TCRBV06-01). Since the ImmunoSeq naming of TCR V genes differs from the IMGT nomenclature used for metaclone definitions, V gene names were made compatible prior to searching for metaclone matches. This included, within the ImmunoSeq annotations, replacement of TCRBV with TRBV, and the removal of leading zeroes from V gene alleles (e.g., replacing TRBV06-06 with TRBV6-6). β-chain data were available for the lung dataset ($n = 13$ TB patients and $n = 3$ cancer controls), the blood dataset ($n = 11$ TB patients and $n = 4$ cancer controls), and the CD4 T cell dataset ($n = 5$ TB patients). All samples per patient were included, and data integrated across disease and tissue groups, resulting in $n = 1,615,131$ TB-associated and $n = 218,372$ cancer-associated β TCRs for the lung dataset; $n = 1,081,593$ TB-associated and $n = 735,834$ cancer-associated β TCRs for the blood dataset; and $n = 336,787$ lung-derived and $n = 219,541$ blood-derived β TCRs for the CD4 T cell dataset.

### Statistics and data visualisation

Analyses were performed in R (version 4.3.3) or Python (version 3.10.4). Data were visualised and figures assembled using R's tidyverse (v2.0.0) and ggpubr (v0.6.0) packages, and Inkscape (v0.92). Statistical differences were assessed using the tests and significance thresholds stated in the text and figure legends. Wilcoxon tests with FDR correction for multiple testing were performed with the wilcox_test or pairwise_wilcox_test functions from the rstatix (v0.7.2) package in R. Base R functions cor() and lm() were used for Spearman correlation and linear regression analyses, respectively, with confint() to calculate confidence intervals for regression coefficients. R packages pheatmap (v1.0.12) and ComplexHeatmap (v2.18.0) were used to create heatmaps. Odds ratios and their confidence intervals were calculated with the fisher.test() function from the stats (v4.3.3) R package. Metaclones were identified using Metaclonotypist (v1.0; written in Python) and visualised as described above. To visualise overlap between CDR3 sequences, an area-proportional Venn diagram was drawn with DeepVenn (https://arxiv.org/abs/2210.04597).

### Reporting summary

Further information on research design is available in the Nature Portfolio Reporting Summary linked to this article.

## Data availability

All source data for the analyses presented in this study are provided in the Source Data file. The processed RNAseq data generated in this study are available at Array Express with accession number E-MTAB-14687. The raw RNA sequencing data in FASTQ format are available under controlled access to comply with data privacy restrictions. Access can be obtained via the European Genome-Phenome Archive with accession number EGAD50000001208. Data will be shared with investigators whose proposed use is within the scope of participant consent subject to a data access agreement. The processed TCR sequencing data generated in this study are available from UCL's Research Data Repository (https://doi.org/10.5522/04/28049606). The raw TCR sequencing data in FASTQ format are available at NCBI Short Read Archive with accession number PRJNA1208718. Previously published single-cell TCR sequencing data from human lung are available from Gene Expression Omnibus with accession numbers GSE253828 (TB patients) and GSE154826 (lung cancer patients). All other data are available in the article and its Supplementary files or from the corresponding author upon request. Source data are provided with this paper.

## Code availability

Analysis code is available on GitHub at https://github.com/carolinturner/tst_tcr (https://doi.org/10.5281/zenodo.18209647). Metaclonotypist library code is available at https://github.com/qimmuno/metaclonotypist (https://doi.org/10.5281/zenodo.17977729).

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

## Acknowledgements

This work was supported by Wellcome Trust awards to M.N. (207511/Z/17/Z and 306550/Z/23/Z). M.N. and G.P. acknowledge support from NIHR Biomedical Research Funding to University College London Hospitals. A.T.M. acknowledges support by the Royal Free Charity. J.C.K. acknowledges support from NIHR Oxford Biomedical Research Centre. J.J. is funded by the Deutsche Forschungsgemeinschaft (DFG, German Research Foundation, 531855214). We thank Michelle Berin, Zandile Maseko and Kimberlee Gunn for supporting participant recruitment and sampling.

## Author contributions

Conceived and designed the study: C.T., A.T.M., B.C., M.N. Sample and clinical data collection: M.B., R.B.M., S.C., M.L., H.K., S.L., P.O., G.P., A.L., M.N. Laboratory analysis: C.T., J.R., A.C., I.U., G.N., S.B., R.B.M., G.P. Data analysis: C.T., A.T.M., R.S., P.Z., J.J., A.K., F.P., W.S., V.R., G.P., J.K., B.C., M.N. Manuscript preparation: C.T., A.T.M., B.C., M.N. with input from all authors

## Competing interests
