## [Transparent Peer Review file · Nature Communications]

Evolution of the tuberculin skin test reveals generalisable Mtb-reactive T cell metaclones

Corresponding Author: Professor Mahdad Noursadeghi

Version 0:

Reviewer comments:

Reviewer #1

(Remarks to the Author)

The manuscript by Turner et al described the TCR repertoire analysis in TST skin test samples in comparison to blood and in vitro stimulated T-cells. The study is well described and describes some interesting findings.

- In Figure 2A, the authors describe the diversity in the TCR repertoire at day 2 and day 7 post TST and conclude that the repertoire is smaller on day 2 and larger at day 7 perhaps due to clonal expansion of the T-cells at the site of the TST. However, there is no mention of the number of T-cells or TCR sequences derived from the biopsies at either of these time points. As the reduced diversity at day 2 could be the result from decreased frequencies of T-cell recruited into the TST site, this should be taken into the equation and mentioned in the data description. Especially when assessing total numbers of unique clones etc total number of TCR sequences obtained is critical to know.

- In Figure 2B, the authors claim a specific depletion of CMV and EBV specific TCRs at day 7 post TST. However as there is also an expansion of Mtb specific T-cells at this time point, thus there may not be a change in the absolute number of CMV and EBV specific TCRs, but only proportional due to expansion of the other populations. This should be clarified, and ideally related to absolute counts for the CMV and EBV specific TCRs.

- Related to Figure 2C, but also in general, the participants in this study were selected on having a positive QFT result in the absence of aTB, which will be a rather heterogenous group. Is anything known on duration of QFT positivity and thus could data be correlated to recent vs remote infection? Or are absolute QFT results available and can heterogeneity of response at day 7 be correlated to magnitude of QFT response? It seems, looking at the diversity in Fig 2C, but also some of the other data points that donors have considerably different responses to the TST administration and wonder if any would correlate to the more routine clinical classifications. Related, did you measure TST induration in the volunteers before taking the punch biopsy, any relation to the number of TCR sequences retrieved? In individuals with large TST induration, where was the punch taken? In the center or also towards the edges?

Does TST induration correlate with the inflammatory and proliferation transcriptomic profiles as depicted in Figure 1?

- Figure 5A suggest that day 7 TST TCR clones are partially not specific to Mtb, could the authors speculate what is happening in these TST sites? As in Figure 2B they described specific depletion of CMV and EBV specific T-cells, which suggests that there is not simply bystander proliferation of all local T-cells resulting from increased inflammatory cytokines. Do to they expect recruitment of T-cells with other specificities?

- The heterogeneity of the study population, with potentially large variation in time since Mtb infection should be included as limitation in the discussion

(Remarks on code availability)

Reviewer #2

(Remarks to the Author)

In this study, the authors analyze gene expression and the TCR repertoire from bulk RNA isolated at days 2 and 7 following intradermal PPD placement, with saline serving as a control for gene expression analyses. They report that day 7 responses are enriched for gene modules associated with cell proliferation, pan-T cell, CD4⁺ T cell, and NK cell activation, but not CD8⁺ T cell responses. TCR repertoire analysis from the same bulk RNA shows that day 2 responses contain more TCRβ clonotypes with ≥2 copies than whole blood, suggesting recruitment of clonally expanded memory T cells to the PPD site.

By day 7, clonal expansion increases further, with reduced diversity reflected in Shannon and inverse Simpson indices. Using CDR3 sequences from three published TB studies, the authors define "Mtb-reactive" TCRs and demonstrate their enrichment in day 7, but not day 2, skin biopsy samples. Similarly, TCRs expanded in PPD-stimulated autologous PBMCs were enriched in day 7, but not day 2, skin biopsies. The absence of enrichment at day 2—the time point typically used clinically to assess the PPD response—is noteworthy and unexpected.

The authors also evaluate TCR clustering methods, proposing a new workflow - metaclonotypist, ultimately selecting TCRdist3 grouping with a well-rationalized threshold of 15. They observe enrichment of "public" metaclones in day 7 PPD biopsies, alongside additional clonotypes lacking clear Mtb specificity, leading to the conclusion that day 7 responses comprise a mix of Mtb-reactive (both private and public) and non-Mtb-specific TCRs. Finally, the manuscript explores HLA associations and enrichment of Mtb-reactive TCRs using GLIPH2 (patterns present in >4 individuals), though important methodological details regarding benchmarking and comparisons with GLIPH2 lacking.

Overall, this manuscript provides an innovative analysis of T cell repertoires 2 and 7 days after PPD placement in LTBI individuals, integrating clustering algorithms and antigen-specific annotations. Nonetheless, several weaknesses remain, as detailed in the comments below.

Major Comments

- It is not clear how clonal frequency was calculated since TCRA and TCRB sequences were reverse-transcribed from bulk RNA isolated from biopsy sites or whole blood. Transcription levels fluctuate with activation state and cell cycle, so repertoire measurements can be biased toward recently activated clones. (e.g., one activated cell with high TCR mRNA might look "expanded"). These issues should at least be discussed, and it should be acknowledged that the interpretation of TCR repertoire using bulk RNA (instead of genomic DNA) is semi-quantitative.
- How do the authors explain that "Mtb-reactive" TCRs are only enriched in d7 but not d2 samples, when they also argue that d2 skin bx samples likely reflect recruitment but not yet proliferation of Mtb-specific T cells? The number of clonotypes present in >1 copies is increased at d2, indicating memory T cells are responding to PPD. Even if they are not significantly expanded, one would expect to observe Mtb-reactive TCR clonotypes at d2. Could this be an issue with the sensitivity of the technique or approach of using bulk RNA from biopsy samples for the purpose of TCR sequencing, which is overcome at the d7 time point by significantly greater clonal expansion of the Mtb-specific T cells? Or that a comparison with the PBMC TCR repertoire is not a fair comparison with skin biopsies using bulk RNA? Otherwise it would appear that the authors are claiming the entire PPD response at day 2 is likely not Mtb specific which seems unlikely since PPD responses are usually negative (ie no induration) in people without LTBI. This issue, and the limitations of the technique should at least be discussed and clarified since it is not consistent with the Discussion point made in lines 294-295.
- The authors also compare the metaclonotypist TCR grouping approach to GLIPH2, including the ability to associate HLA restriction (lines 226-229). However, as previously published (e.g. Musvosvi et al., Nat Med, 2023), GLIPH2 generates large numbers of motifs / metaclones some of which are ultimately removed by statistical thresholds in Fisher Exact Test, CDR3 Length Score, Vb usage score, HLA association score, etc). There is no description of which tests (and their p value thresholds) were used to determine the final list of GLIPH2 patterns (and their TCRs) which were compared with the metaclones. Therefore, the comparison could be invalid. The comparison should also be depicted in one of the main figures since validation of the metaclonotypist approach is a main element of the manuscript yet is not demonstrated in any figures—only an Extended Datasheet is referenced.
- There are many statements about data made in lines 242-252 without any figure call-outs or references. Could the authors support these statements with figures since they don't all seem to be related to Figure 5A.
- The HLA association for GLIPH2 does not always find just a single association, but often 2. This is especially true for HLA alleles in linkage disequilibrium (ie. HLA DRB1*07:01 and DRB4*01:03) since it is usually impossible to distinguish which DR allele is the restricting MHC for such alleles. However, only a single association is listed for these metaclones listed. Could the authors re-examine the HLA associations to comprehensively report the HLA associations or elaborate on how single HLA associations were made for all TCRs in this dataset?
- For the large amount of data analyzed, the main figures are limited to summary statistics and group frequency comparisons. This seems like a missed opportunity. There should at least be a presentation of the most meaningful Mtb-specific, HLA-restricted metaclones in the main figures and the antigen specificities for any that have been already defined in the literature (using VDJDDB or IEDB).
- Several publications have reported responses by donor-unrestricted T (DURT) cells to Mtb. The dataset generated in this paper allows analysis of whether any of the responding T cells in the Mtb-specific subset (infected vs peptide-pulsed APCs) include TCR-alpha sequences that match those of canonical MR1-restricted T cells, or CD1B, CD1c and CD1d-restricted T cells. I would expect low enrichment at both the d2 and d7 time points (compared to PBMCs) since intradermal PPD was used.

Minor Comments

- In line 258, the context of the "metaclones" is unclear. Are the authors referring to Mtb-specific TCRs?
- The authors frequently refer to public and private metaclones (especially lines 253-275), public T cell responses, and TCRs throughout the manuscript without defining this terminology. Perhaps it simply needs to be made consistent throughout the manuscript and clearly defined in the Results Section. Grouping algorithms are used to group what would be considered "private" TCRs (unique to individuals) that contain the same central motifs as other private TCRs, and therefore usually recognize the same antigens in the context of the same MHC molecules. However, the TCRs, themselves, are not necessarily public.
- It is not clear what is meant by the statement in lines 246-247. If broader metaclones were not enriched, then why would it be more likely that the more specific / less broad metaclones would be?
- In lines 270-272 the authors state, "Despite the dominance of private TCRs at individual level, evaluation of the cumulative

publicity of metaclones ranked by their publicity suggests that as few as 10 metaclones are sufficient to identify Mtb-T cell reactivity in the entire study population". It is not clear to me what this is intended to convey. All individuals would be expected to have mostly private and some public TCRs in any immune response. Do the authors suggest a threshold for the number of metaclones (e.g. 10) present in a PPD response that would determine whether that response was Mtb-specific? Is there reason to think any of the responses do not contain at least some Mtb-specific TCRs despite the fact that all participants were reportedly IGRA+?

(Remarks on code availability)

Reviewer #3

(Remarks to the Author)

In this manuscript Turner et al. take a unique approach to determine T cell recruitment and clonality into a temporal antigen challenge site. Analysis at two timepoints (day 2 and 7) following antigen exposure provides beneficial insights into the timing and potential kinetics of T cell recruitment. Turner et al. found initial recruitment during the peak inflammatory response was predominantly non-antigen specific, as determined by no enrichment for Mtb-specific CDR3s compared to blood. These T cells were then replaced by day 7 with predominantly expanded Mtb-reactive T cells. Turner et al. developed a modular computational pipeline to identify public alpha or beta TCR sequences, establishing of a catalogue of public Mtb-specific HLA-restricted T cell sequences. While the majority of Mtb-reactive T cells were private, computational analysis revealed 10 metaclones (single chain only), suggesting population-level immunodominance of Mtb-specific responses.

All analysis seems to be performed in samples from healthy individuals with evidence of peripheral blood Mtb-reactive T cells. Do the authors believe the preexisting Mtb-reactive T cells in the peripheral blood were naïve T cells or the result of prior exposure?

While examining T cell responses at the site of antigen challenge over peripheral blood is commendable, it is worth noting that this is not the site of infection in TB. The authors should acknowledge that recruitment of T cell into the skin during a controlled antigenic challenge may be remarkably different to recruitment of T cells into sites like the lung during infection.

Figure 3B, does each column across the x-axis represent one participant?

Which figure shows the lower enrichment of the full set of day 7 TST TCR clones compared to the metaclones (page 8, line 249)?

One limitation of this bulk sequencing approach, which the authors acknowledge, is the lack of paired alpha/beta-chain data, with the public metaclones identified being restricted to a beta-chain only. I recommended the authors are careful in the language about generalizing immunodominance to specific peptides (line 292) as the metaclones identified are only beta-chains and therefore insufficient to pursue reverse epitope discovery.

The work appears to be novel and supports the conclusions of the authors, providing interesting data regarding the timing, clonality and publicness of T cell recruitment into an antigenic site. However, without paired alpha/beta TCR data, the applicability of these findings to disease stratification and epitope discovery appears to be limited.

(Remarks on code availability)

Version 1:

Reviewer comments:

Reviewer #1

(Remarks to the Author)

Thanks to the authors for replying to my comments so extensively, including additional analysis. I think these extra data have improved the manuscript and clarified my initial concerns on TCR numbers.

(Remarks on code availability)

Reviewer #2

(Remarks to the Author)

All of my comments have been addressed.

(Remarks on code availability)

Reviewer #3

(Remarks to the Author)

Thank you for the revised manuscript and detailed response letter. While the majority of my comments have been addressed and the manuscript provides new insights into TCR recruitment to a temporal antigen challenge site, without paired TCR data or experimental validation of HLA restriction/public clones, the applicability of these findings as a resource for biomarker discovery and reverse epitope discovery is limited and the work likely does not significantly advance the field.

Outside of this, my only other suggestion is to include the additional details about the participant cohort in the results section of the paper (line 90). The authors response to Reviewer #1, that they are healthy volunteers with known exposures, was a helpful qualification and I believe the manuscript will be strengthened with more upfront inclusion of this information.

(Remarks on code availability)

Response to reviewers

As requested, we have provided point by point responses to the reviewers' comments and indicated the changes we have made below.

Reviewer #1

1. *The manuscript by Turner et al described the TCR repertoire analysis in TST skin test samples in comparison to blood and in vitro stimulated T-cells. The study is well described and describes some interesting findings.*

We thank the reviewer for the positive comment.

2. *In Figure 2A, the authors describe the diversity in the TCR repertoire at day 2 and day 7 post TST and conclude that the repertoire is smaller on day 2 and larger at day 7 perhaps due to clonal expansion of the T-cells at the site of the TST. However, there is no mention of the number of T-cells or TCR sequences derived from the biopsies at either of these time points. As the reduced diversity at day 2 could be the result from decreased frequencies of T-cell recruited into the TST site, this should be taken into the equation and mentioned in the data description. Especially when assessing total numbers of unique clones etc total number of TCR sequences obtained is critical to know.*

The reviewer is correct in saying that repertoire diversity metrics are affected by the total number of TCR sequences. We address this by down-sampling the repertoires to the same number of TCR sequences as stated in line 127 and the legend for Figure 2. For clarification, we have now added summary statistics for the total number of TCR sequences obtained for each sample type and include the total number of TCR sequences to which each repertoire was down-sampled in the Results text (lines 121-125).

"The median number of total β -chain TCRs obtained was 267,057 (range 113-566,403) for day 7 TSTs; 53,756 (range 2,596-100,381) for day 2 TSTs; and 66,452 (range 18,266-119,984) for peripheral blood samples. Since repertoire diversity metrics are affected by sequencing depth (Supplementary Figure 3B-C), we excluded samples with very small repertoires and down-sampled repertoires to the same size (n=16,000 total TCRs) prior to this analysis."

Importantly, day 7 TSTs have significantly lower diversity and a higher frequency of expanded TCRs (with count >1) supporting our conclusion that there is oligoclonal T cell expansion between day 2 and day 7.

3. *In Figure 2B, the authors claim a specific depletion of CMV and EBV specific TCRs at day 7 post TST. However as there is also an expansion of Mtb specific T-cells at this time point, thus there may not be a change in the absolute number of CMV and EBV specific TCRs, but only proportional due to expansion of the other populations. This should be clarified, and ideally related to absolute counts for the CMV and EBV specific TCRs.*

We agree with the reviewer that the expansion of Mtb-reactive T cells might drive the apparent depletion of T cells of other specificities from the day 7 TST repertoire. We have amended the Results text in line with this possible interpretation (lines 148-150):

"Day 7 TSTs showed a statistically significant reduction in the relative frequencies of both CMV and EBV-reactive CDR3 sequences compared to blood and day 2 TSTs, consistent with larger clonal expansions of Mtb-reactive sequences."

We have also amended our Discussion (lines 324-328) to discuss this point explicitly as follows (see also response to comment 6):

"Our data cannot fully discriminate the extent to which reduction of non-Mtb-reactive T cells at the TST site by day 7 is due to their clearance or their relative decrease caused by the expansion of Mtb-reactive TCRs dominating the sampled repertoire. The latter process is supported by our observation that the full set of day 7 TST TCRs shows substantially lower enrichment in independent TB-associated datasets compared to day 7 TST metaclones."

The analysis in Figure 2B was performed in equal-sized repertoires. Therefore, depicting the absolute count of CMV and EBV specific CDR3 sequences, instead of their percentage, would have no impact on the data or their interpretation.

Interpretation of absolute counts of published CMV-, EBV- and Mtb-reactive TCRs in the full repertoires is confounded by the large technical variability in the number of recovered TCR sequences per sample. However, for the reviewer's interest, we have calculated these numbers for the beta chain repertoire as follows:

	Blood (n=20)	TST_D2 (n=17)	TST_D7 (n=165)
All TCRs			
CMV-reactive CDR3s	Median: 546 Range: 221-1436	Median: 601 Range: 21-1584	Median: 574 Range: 2-6598
EBV-reactive CDR3s	Median: 190.5 Range: 63-1687	Median: 202 Range: 9-1499	Median: 225 Range: 2-3578
Mtb-reactive CDR3s	Median: 898.5 Range: 265-2080	Median: 743 Range: 35-3964	Median: 2633 Range: 3-34430
Expanded TCRs			
CMV-reactive CDR3s	Median: 240 Range: 57-564	Median: 368 Range: 21-1375	Median: 500 Range: 0-6352
EBV-reactive CDR3s	Median: 87 Range: 13-1588	Median: 148 Range: 9-1441	Median: 199 Range: 0-3411
Mtb-reactive CDR3s	Median: 366 Range: 83-948	Median: 485 Range: 33-3674	Median: 2430 Range: 0-33995

4. *A. Related to Figure 2C, but also in general, the participants in this study were selected on having a positive QFT result in the absence of aTB, which will be a rather heterogenous group. Is anything known on duration of QFT positivity and thus could data be correlated to recent vs remote infection? Or are absolute QFT results available and can heterogeneity of response at day 7 be correlated to magnitude of QFT response? It seems, looking at the diversity in Fig 2C, but also some of the other data points that donors have considerably different responses to the TST administration and wonder if any would correlate to the more routine clinical classifications.*

No data are available for duration of QFT positivity or absolute QFT results. However, we can stratify participants into presumed 'recent' and 'remote' exposure groups, based on epidemiological criteria. We have included an additional Supplementary Figure (Supplementary Figure 15) and amended the manuscript to reflect these additional analyses.

Methods lines 423-426: "In a sensitivity analysis, participants were stratified by the indication for QuantiFERON testing as surrogate for possible interval time from exposure. In this context, household contacts of active TB index cases were expected to be enriched for individuals with more recent exposures than individuals being tested via routine occupational health screening."

Results lines 292-295: "Finally, we stratified participants into groups that were expected to be enriched for either 'recent' or 'remote' exposures, based on epidemiological criteria. We found a modest increase in TST induration in the 'recent exposure' group, but found no differences for any of the TST metrics measured on the molecular level (Supplementary Figure 15)."

To summarise, no differences were found between household contacts of TB index cases (enriched for recent exposures) and occupational health screening participants (enriched for remote exposures) in terms of a) the extent of inflammation in the TST measured by transcriptional modules, b) TCR sequence diversity or co-incidence metrics in the TST on day 7, and c) abundance of Mtb-reactive TCRs in the day 7 TSTs (measured by public Mtb-reactive CDR3s, percentage Mtb-reactive metaclones, and percentage of expanded TCRs).

Supplementary Figure 15: Comparison of TST metrics between participants with presumed recent vs. remote exposure to TB. Household contacts (HC) of active TB index cases were defined as recently exposed to TB, and participants whose immunological memory to Mtb was identified via routine occupational health (OH) screening were defined as remotely exposed to TB. **(A)** Day 2 TST induration (n=27 HC vs. n=86 OH). **(B)** Expression of cellular proliferation and cell type-specific modules in Day 2 and Day 7 TST bulk RNAseq data (D2: n=30 HC vs. n=100 OH; D7: n=27 HC vs. n=67 OH). **(C-E)** Diversity metrics **(C)**, within donor coincidence **(D)** and between donor coincidence **(E)** in down-sampled Day 7 TST bulk TCRseq beta chain data (n=17 HC vs. n=43 OH, resulting in n=136 pairwise HC and n=903 pairwise OC comparisons to calculate between donor coincidence). **(F-H)** Proxies for Mtb-reactivity in down-sampled Day 7 TST bulk TCRseq beta chain data (n=17 HC vs. n=43 OH), including percentage of public Mtb-reactive CDR3 sequences **(F)** and metaclone TCRs **(G)** amongst all or expanded (clone size >1) TCR sequences, and percentage of expanded TCRs, stratified by clone size (>1, >2, >3, >4) **(H)**. Statistical significance was assessed with Wilcoxon tests.

B. Related, did you measure TST induration in the volunteers before taking the punch biopsy, any relation to the number of TCR sequences retrieved? In individuals with large TST induration, where was the punch taken? In the center or also towards the edges?

The reviewer raises the question whether clinically measurable TST induration on day 2 is associated with the extent of T cell reactivity as measured at the sequence level. The number of retrieved TCR sequences varies substantially in our dataset (see response to comment 2) and is likely not only determined by T cell density in the skin biopsy but also by technical factors such as RNA quality. Instead of showing correlation between TST induration and absolute number of recovered TCR sequences, we therefore explored the relation between day 2 TST induration and percentage of metaclones or percentage of expanded TCRs, as proxies for the number of Mtb-reactive TCRs in the day 7 TST. This shows a statistically significant, positive correlation of D2 TST induration with D7 TST metaclone proportion and expanded D7 TST TCRs (clone size >3), suggesting that both are the result of a common determinant of the TST response. Inflammatory induration recedes by day 7, therefore it is not possible to assess the relationship between induration and Mtb-reactive T cell frequencies at this time point.

We have included this observation and our interpretation of it in the Results and Discussion sections:

Lines 271-274: “In contrast, at day 7 metaclones were about 10-fold enriched in the TST samples compared to blood and day 2 TST repertoires (Figure 5B, Supplementary Figure 8), and the percentage of day 7 TST repertoires captured by metaclones was positively correlated with the TST induration measured at day 2 (Supplementary Table 1).”

Lines 337-341: “We also observed a positive correlation between TST induration on day 2 and the extent of predicted Mtb reactivity on day 7 (measured as proportion of metaclones, or proportion of expanded TCRs), consistent with the hypothesis that both arise from a common determinant of the TST response, likely to be the frequency of circulating Mtb-reactive memory T cells that initially drive inflammatory induration and then proliferate in response to antigen.”

	Day 2 TST	Day 7 TST	Blood
Transcriptional modules			
Module ‘All T cells’	R=0.32, p<0.001	R=0.32, p<0.001	NA
Module ‘CD4 T cells’	R=0.31, p<0.001	R=0.4, p<0.001	NA
Module ‘CD8 T cells’	R=0.47, p<0.001	R=0.22, p<0.001	NA
Module ‘NK cells’	R=0.15, p=0.043	R=-0.03, p=0.732	NA
Module ‘Myeloid cells’	R=0.45, p<0.001	R=0.12, p=0.174	NA
Module ‘Proliferation’	R=-0.21, p=0.005	R=0.064, p=0.458	NA
TCR metrics			
% metaclones	R=0.83, p=0.058	R=0.22, p=0.021	R=0.018, p=0.969
% TCRs with clone size >1	R=-0.43, p=0.419	R=0.061, p=0.525	R=0.71, p=0.088
% TCRs with clone size >2	R=-0.14, p=0.803	R=0.18, p=0.064	R=0.61, p=0.167
% TCRs with clone size >3	R=0.14, p=0.803	R=0.26, p=0.007	R=0.61, p=0.167
% TCRs with clone size >4	R=0.14, p=0.803	R=0.31, p<0.001	R=0.61, p=0.167

Supplementary Table 1: Day 2 TST induration is correlated with T cell gene signatures and proxies for Mtb reactivity in day 7 TSTs. Spearman correlation analysis of TST induration on day 2 with expression of gene signatures in TSTs from day 2 (n=184) or day 7 (n=135), and with percentage of β -chain metaclone TCRs or percentage of expanded TCRs (stratified by clone size) in down-sampled repertoires from day 2 TST (n=6), day 7 TST (n=111) and blood (n=7). Significant correlations (p<0.05) are highlighted in bold.

In addition, we confirm that the punch biopsies were taken in the centre of the TST induration, and we have now included this in the methods (lines 418-419): “The precise injection sites were marked with permanent marker pen and used to position the site of the punch biopsies at designated time points.”

5. Does TST induration correlate with the inflammatory and proliferation transcriptomic profiles as depicted in Figure 1?

We include the correlation analysis between day 2 TST induration and transcriptional modules in an additional supplementary figure (see also response to comment 4). We reference this in the Results section (lines 111-113): “Clinical induration at TST sites on day 2 was positively correlated with multiple cell types at day 2, but only T cell accumulation at day 7, as measured by cell-type specific gene expression of modules (Supplementary Table 1).”

6. *Figure 5A suggest that day 7 TST TCR clones are partially not specific to Mtb, could the authors speculate what is happening in these TST sites? As in Figure 2B they described specific depletion of CMV and EBV specific T-cells, which suggests that there is not simply bystander proliferation of all local T-cells resulting from increased inflammatory cytokines. Do they expect recruitment of T-cells with other specificities?*

We have amended the Discussion (lines 320-328) to clarify our interpretation of the data: “Recruitment appears to be non-selective, demonstrated by the similar proportions of Mtb-, tetanus toxoid-, CMV- and EBV-reactive CDR3 sequences observed in the day 2 TSTs and blood samples. However, Mtb-specific sequences are enriched relative to CMV-, EBV- and tetanus toxoid-reactive CDR3 sequences in day 7 TSTs, suggesting that the proliferation response is not simply due to bystander proliferation of all local T cells. Our data cannot fully discriminate the extent to which reduction of non-Mtb-reactive T cells at the TST site by day 7 is due to their clearance or their relative decrease caused by the expansion of Mtb-reactive TCRs dominating the sampled repertoire. The latter process is supported by our observation that the full set of day 7 TST TCRs shows substantially lower enrichment in independent tuberculosis-associated datasets compared to day 7 TST metaclones.”

7. *The heterogeneity of the study population, with potentially large variation in time since Mtb infection should be included as limitation in the discussion*

We have added the following text to the Discussion (lines 358-363): “Inter-individual variation in total T cell reactivity and/or the clonal repertoire, including intra- and inter-donor co-incidence metrics in the TST may relate to duration of Mtb infection or time interval since immunological clearance of bacteria. While neither of these variables can be measured accurately, epidemiological classification of our study cohort into household contacts, likely to be enriched for recent exposures, and occupational health screening participants, likely to be enriched for remote exposures, showed no differences in any of the measured molecular TST metrics.”

Reviewer #2

8. *In this study, the authors analyze gene expression and the TCR repertoire from bulk RNA isolated at days 2 and 7 following intradermal PPD placement, with saline serving as a control for gene expression analyses. They report that day 7 responses are enriched for gene modules associated with cell proliferation, pan-T cell, CD4⁺ T cell, and NK cell activation, but not CD8⁺ T cell responses. TCR repertoire analysis from the same bulk RNA shows that day 2 responses contain more TCRβ clonotypes with ≥2 copies than whole blood, suggesting recruitment of clonally expanded memory T cells to the PPD site. By day 7, clonal expansion increases further, with reduced diversity reflected in Shannon and inverse Simpson indices. Using CDR3 sequences from three published TB studies, the authors define “Mtb-reactive” TCRs and demonstrate their enrichment in day 7, but not day 2, skin biopsy samples. Similarly, TCRs expanded in PPD-stimulated autologous PBMCs were enriched in day 7, but not day 2, skin biopsies. The absence of enrichment at day 2—the time point typically used clinically to assess the PPD response—is noteworthy and unexpected. The authors also evaluate TCR clustering methods, proposing a new workflow - metaclonotypist, ultimately selecting TCRdist3 grouping with a well-rationalized threshold of 15. They observe enrichment of “public” metaclones in day 7 PPD biopsies, alongside additional clonotypes lacking clear Mtb specificity, leading to the conclusion that day 7 responses comprise a mix of Mtb-reactive (both private and public) and non-Mtb-specific TCRs. Finally, the manuscript explores HLA associations and enrichment of Mtb-reactive TCRs using GLIPH2 (patterns present in >4 individuals), though important methodological details regarding benchmarking and comparisons with GLIPH2 lacking. Overall, this manuscript provides an innovative analysis of T cell repertoires 2 and 7 days after PPD placement in LTBI individuals, integrating clustering algorithms and antigen-specific annotations. Nonetheless, several weaknesses remain, as detailed in the comments below.*

We thank the reviewer for this summary.

9. *It is not clear how clonal frequency was calculated since TCRA and TCRB sequences were reverse-transcribed from bulk RNA isolated from biopsy sites or whole blood. Transcription levels fluctuate*

with activation state and cell cycle, so repertoire measurements can be biased toward recently activated clones. (e.g., one activated cell with high TCR mRNA might look “expanded”). These issues should at least be discussed, and it should be acknowledged that the interpretation of TCR repertoire using bulk RNA (instead of genomic DNA) is semi-quantitative.

The reviewer is correct that the UMI-based method quantifies the number of mRNA transcripts, not the number of T cells. However, unlike B cells, T cells do not substantially change mRNA levels upon activation (see Oakes et al. Quantitative Characterization of the T Cell Receptor Repertoire of Naïve and Memory Subsets Using an Integrated Experimental and Computational Pipeline Which Is Robust, Economical, and Versatile. *Front Immunol.* 2017 Oct 12;8:1267. doi: 10.3389/fimmu.2017.01267. PMID: 29075258; PMCID: PMC5643411. and references therein). In fact, there is a small downregulation of TCR mRNA upon activation. But the changes are in the order of 2-fold maximum, which is within the error range of the PCR assay itself. We therefore are confident that the assay gives a good estimate of clonal expansion.

We have edited the text to clarify this point (lines 473-476): “Since the UMI-based method quantifies the number of mRNA transcripts, we note that ‘clone size’ does not measure the number of T cells directly. Nevertheless, mRNA number is a good proxy for clone size and clonal expansion since T cells do not substantially change TCR mRNA levels upon activation²⁶.”

10. How do the authors explain that “Mtb-reactive” TCRs are only enriched in d7 but not d2 samples, when they also argue that d2 skin bx samples likely reflect recruitment but not yet proliferation of Mtb-specific T cells? The number of clonotypes present in >1 copies is increased at d2, indicating memory T cells are responding to PPD. Even if they are not significantly expanded, one would expect to observe Mtb-reactive TCR clonotypes at d2. Could this be an issue with the sensitivity of the technique or approach of using bulk RNA from biopsy samples for the purpose of TCR sequencing, which is overcome at the d7 time point by significantly greater clonal expansion of the Mtb-specific T cells? Or that a comparison with the PBMC TCR repertoire is not a fair comparison with skin biopsies using bulk RNA? Otherwise it would appear that the authors are claiming the entire PPD response at day 2 is likely not Mtb specific which seems unlikely since PPD responses are usually negative (ie no induration) in people without LTBI. This issue, and the limitations of the technique should at least be discussed and clarified since it is not consistent with the Discussion point made in lines 294-295.

The reviewer raises an important point which we already discuss (lines 330-337) including reference to the limit of sensitivity to detect Mtb-reactive sequences by enrichment compared to blood. Since there is no detectable proliferation at day 2, T cell accumulation in the TST at this time point must represent recruitment from the circulation (lines 318-319), therefore inflammatory reactivity at day 2 most likely reflects frequency of circulating Mtb-reactive T cells, irrespective of the fact that they may not be enriched in the recruited T cell repertoire relative to the blood of individuals with prior exposure and memory.

We have now clarified this both in the Results (line 143-145):

“This indicates that the early inflammatory response likely reflects the circulating frequency of Mtb-specific T cells in individuals with prior memory, rather than preferential recruitment of specific clones.”

And in the Discussion (lines 337-341): “We also observed a positive correlation between TST induration on day 2 and the extent of predicted Mtb reactivity on day 7 (measured as proportion of metaclones, or proportion of expanded TCRs), consistent with the hypothesis that both arise from a common determinant of the TST response, likely to be the frequency of circulating Mtb-reactive memory T cells that initially drive inflammatory induration and then proliferate in response to antigen.”

11. The authors also compare the metaclonotypist TCR grouping approach to GLIPH2, including the ability to associate HLA restriction (lines 226-229). However, as previously published (e.g. Musvosvi et al., Nat Med, 2023), GLIPH2 generates large numbers of motifs / metaclones some of which are ultimately removed by statistical thresholds in Fisher Exact Test, CDR3 Length Score, Vb usage score, HLA association score, etc). There is no description of which tests (and their p value thresholds) were used to determine the final list of GLIPH2 patterns (and their TCRs) which were compared with the metaclones. Therefore, the comparison could be invalid. The comparison should also be depicted in one of the main figures since validation of the metaclonotypist approach is a main element of the manuscript yet is not demonstrated in any figures- only an Extended Datasheet is referenced.

We agree with the reviewer that it is difficult to decide at what stage of a TCR grouping pipeline comparisons should be made. This consideration motivated us to focus the main text figures on rigorous

benchmarking of clustering algorithms using TCRs with known specificity, to identify Pareto-optimal approaches (Fig. 4B), and on demonstrating false discovery control through HLA shuffling (Fig. 4C). We believe these analyses represent a key advance over prior studies comparing clustering tools, which often yielded inconclusive results as they did not isolate performance differences related to distinct stages of the computational pipelines.

As the reviewer correctly points out, the GLIPH2 output can be filtered by different criteria, thus potentially increasing specificity. In our primary analysis, we mirrored the Metaclonotypist approach and filtered raw GLIPH2 clusters by publicity (containing TCRs from at least 4 different individuals) and HLA association (Fisher's Exact Test with $FDR < 0.1$), as already described in lines 583-586. This choice of filtering strategy ensured a controlled comparison of algorithmic approaches at equivalent levels of clustering specificity (Figure 5A). In a new main text figure panel (Figure 4F), we now visualise the overlap of clustered unique CDR3 amino acid sequences between Metaclonotypist and GLIPH2, to highlight the increased sensitivity of our approach.

In response to the reviewers question, the revised manuscript furthermore reports additional analyses (in new Supplementary Figure 10), which demonstrate the effect of applying filters to GLIPH2 output as described by Musvosvi et al., either alone or in combination with the HLA association test at $p < 0.05$ (as described by Musvosvi et al.) or at $FDR < 0.1$ (as used in the Metaclonotypist pipeline). These analyses demonstrate that specificity (enrichment of metaclones in independent data associated with TB) is markedly improved by the Metaclonotypist approach of controlling HLA associations for multiple testing, irrespective of additional filtering, such as for CDR3 length or Vb usage score.

We have amended the manuscript as follows:

Results, lines 234-242:

"After exploring different GLIPH2 filters (Supplementary Figure 10A), we retained clusters which passed the same stringent HLA association test with multiple testing correction applied in the Metaclonotypist pipeline. These GLIPH2 clusters (Supplementary File S7-8) also primarily associated with class II alleles, which demonstrates the robustness of our findings with respect to the algorithmic approach. However, Metaclonotypist clustered almost three times as many unique CDR3s compared to GLIPH2, suggesting that Metaclonotypist offers increased sensitivity to detect metaclone associations (Figure 4F). Indeed, comparison of CDR3s clustered by Metaclonotypist or GLIPH2 with VDJdb class II-restricted β -chain CDR3 entries annotated as Mtb-reactive found three matches in the GLIPH2 output, but an additional match in the Metaclonotypist output (Figure 4F, Supplementary Figure 9A)."

Results, lines 257-260:

"Metaclonotypist motifs showed comparable enrichment to CDR3 β sequences from GLIPH2 clusters with statistically significant class II HLA-allele associations (Figure 5A; blue vs. red) suggesting that Metaclonotypist retains similarly high specificity, despite clustering a larger proportion of TCRs into metaclones (Figure 4F). Additional GLIPH2 filters did not improve performance (Supplementary Figure 10B)."

Methods, lines 583-590:

"GLIPH2 analysis was undertaken on the same set of Day 7 TST β -chain repertoires as described for metaclone discovery above, using default settings and CD48_v2.0 reference. The Metaclonotypist approach was then mirrored, by selecting only GLIPH2 similarity clusters containing TCRs from ≥ 4 individuals and testing for associations with HLAs found in ≥ 4 individuals (done separately for class I and class II alleles). We explored the effect of filtering the GLIPH2 output further, as described before¹⁵. Filter 1 selected clusters that consisted of ≥ 3 unique CDR3s and had a Fisher_score, vb_score and length_score ≤ 0.05 each. Filter 2 applied the Fisher's exact test for HLA association, either with a significance threshold of $p < 0.05$ (as used by Musvosvi et al¹⁵) or with an $FDR < 0.1$ as applied in the Metaclonotypist pipeline."

CDR3s in metaclones

Figure 4F. Venn diagram showing the overlap of unique β chain CDR3 amino acid sequences included in class II-associated metaclone clusters by Metaclonotypist or GLIPH2 and annotated as class 2-restricted Mtb-reactive TCRs in VDJDdb.

A

	Number clusters	Number unique CDR3s
Gliph2 + Filter 1	1837	6546
Gliph2 + Filter 1 + Filter 2 ($p < 0.05$)	1342	5022
Gliph2 + Filter 1 + Filter 2 (FDR < 0.1)	106	476
Gliph2 + Filter 2 (FDR < 0.1)	128	476

B

Supplementary Figure 10: Effect of filtering on sensitivity and specificity of Gliph2 similarity clusters. Gliph2 similarity clusters were derived from expanded (TCR count > 1) day 7 TST β -chain repertoires (122,253 TCR clones, 151 individuals). Gliph2 clusters containing CDR3s from ≥ 4 participants were selected for analysis. Filter 1 retained clusters with ≥ 3 unique CDR3s, which had a Fisher_score, vb_score and length_score ≤ 0.05 each. Filter 2 retained clusters with a Fisher's exact test for HLA class II association of $p < 0.05$ or FDR < 0.1 . **A.** Number of Gliph2 clusters and number of unique day 7 TST CDR3s selected in each analysis iteration. **B.** Relative enrichment of CDR3s clustered by Gliph2 and filtered by different criteria in multiple external data sets, showing point estimates and 95% confidence intervals of odds ratios in the pairwise comparisons indicated. See Figure 5A in the main manuscript for a detailed description of the datasets. Results for 'Gliph2 + Filter 2 (FDR < 0.1)' are replicated in Figure 5A in the main manuscript as 'Gliph2-matching CDR3s'.

12. There are many statements about data made in lines 242-252 without any figure call-outs or references. Could the authors support these statements with figures since they don't all seem to be related to Figure 5A.

All the text in this section does in fact relate to Figure 5A. We have included specific references to Figure 5A to guide readers more clearly through this section (lines 255-266):

"We benchmarked this analysis for publicity and Mtb-reactivity of metaclones identified using Metaclonotypist against metaclones identified by the GLIPH2 algorithm. Metaclonotypist motifs showed

comparable enrichment to CDR3 β sequences from GLIPH2 clusters with statistically significant class II HLA-allele associations (Figure 5A; blue vs. red) suggesting that Metaclonotypist retains similarly high specificity, despite clustering a larger proportion of TCRs into metaclones (Figure 4F). Additional GLIPH2 filters did not improve performance (Supplementary Figure 10). Importantly, amino acid metaclone motifs as defined by GLIPH2 were not substantially enriched in TB samples in these datasets (Figure 5A; light-brown), indicating the need for more restrictive motif definitions. Interestingly, compared to metaclones, we found substantially lower enrichment of the full discovery set of expanded day 7 TST β TCRs (Figure 5A; grey vs. blue), suggesting that a substantial proportion of day 7 TST TCR clones may not be specific to Mtb, and that identification of metaclones significantly improves antigen-agnostic enrichment of the Mtb-reactive T cell response.”

*13. The HLA association for GLIPH2 does not always find just a single association, but often 2. This is especially true for HLA alleles in linkage disequilibrium (ie. HLA DRB1*07:01 and DRB4*01:03) since it is usually impossible to distinguish which DR allele is the restricting MHC for such alleles. However, only a single association is listed for these metaclones listed. Could the authors re-examine the HLA associations to comprehensively report the HLA associations or elaborate on how single HLA associations were made for all TCRs in this dataset?*

We thank the reviewer for prompting us to clarify this. Where more than one significant HLA association is found, we had chosen to report the most significant association, as it represents a likely candidate for the causal association when linkage disequilibrium is not total. For greater clarity, we now refer to these as ‘lead HLA association’ and have amended Supplementary Files 5-8 to include all associations that passed our statistical testing. For metaclones with more than one significant association, the most common combinations are DQA1*05_DQB1*03 and DRB1*11; DQA1*02_DQB1*02 and DRB1*07; and DQA1*05_DQB1*02 and DRB1*03. All represent HLA allele combinations in linkage disequilibrium (Gragert et al., Human Immunology, 2013; <https://doi.org/10.1016/j.humimm.2013.06.025>).

We have amended the discussion (lines 381-387) as follows, to acknowledge this limitation:

“Consistent with most data on HLA class II restriction of CD4 T cell responses, we found the lead HLA associations for day 7 TST metaclones to be predominantly HLA-DR alleles. This skew is attributed to the higher prevalence of diverse DR alleles, a structure that allows them to bind a wider variety of peptides, and potentially higher levels of expression than HLA-DP and HLA-DQ alleles on antigen presenting cells³⁵⁻³⁷. Several metaclones were significantly associated with more than one HLA allele, likely due to the co-expression of HLA alleles in linkage disequilibrium³⁸, limiting the ability to resolve which allele is the restricting MHC for such metaclones.”

14. For the large amount of data analyzed, the main figures are limited to summary statistics and group frequency comparisons. This seems like a missed opportunity. There should at least be a presentation of the most meaningful Mtb-specific, HLA-restricted metaclones in the main figures and the antigen specificities for any that have been already defined in the literature (using VDJDDB or IEDB).

We have included this additional analysis as requested by the reviewer. We compared 1392 CDR3 sequences included in the 177 class 2-restricted metaclones against 30 CDR3 beta sequences, annotated in VDJdb with MHC class "MHCII", and antigen species "M.tuberculosis" or "Mtb".

This resulted in 4 matching CDR3s from 3 different metaclone clusters. We list those matches in Supplementary Figure 9A.

In Figure 4, we already visualise the adjacency graphs and sequence logos for the most public and most abundant metaclones in the Day 7 TST (index 8 and 21). In addition, we now include in Supplementary Figure 9B the adjacency graphs and sequence logos for the most public and most abundant metaclones found in the set of Mtb-reactive T cells by Musvosvi et al. (index 156 and 100), and for metaclones with Mtb CDR3 matches in VDJdb (index 100, 47 and 12).

Furthermore, we now include in Table 3 the metaclone descriptions (including consensus CDR3 motif, V gene usage, and lead HLA association with its pvalue and odds ratio) for the 10 most public metaclones in our down-sampled Day 7 TST dataset, which together identify Mtb-T cell reactivity in the entire study population as demonstrated in Figure 5D.

We reference these additional display items in Results lines 228-230, 238-242, and 286-289:

“The identified single chain metaclones can be represented as sequence motifs and visualised as adjacency graphs in which labelling of the individual TCR sequence by the donor reveals substantial publicity (Figure 4D, Supplementary Figure 9B).”

“However, Metaclonotypist clustered almost three times as many unique CDR3s compared to GLIPH2, suggesting that Metaclonotypist offers increased sensitivity to detect metaclone associations (Figure 4F). Indeed, comparison of CDR3s clustered by Metaclonotypist or GLIPH2 with VDJdb class II-restricted β -chain CDR3 entries annotated as Mtb-reactive found three matches in the GLIPH2 output, but an additional match in the Metaclonotypist output (Figure 4F, Supplementary Figure 9A).”

“Despite the dominance of private PPD-reactive TCRs at individual level, evaluation of the cumulative publicity of metaclones ranked by their publicity suggests that as few as 10 metaclones are sufficient to identify Mtb-T cell reactivity in the entire study population (Figure 5D, Table 3, Supplementary Figure 14).”

A

Metaclone description	Matching CDR3	VDJdb entry
Index 100 Consensus CDR3: CSARGQGNEQFF V: TRBV20-1	CSARGQGNEQYF	V: TRBV20-1*01 Species: M.tuberculosis Protein: PPE33 Epitope: PPQIAANRSQLISLV
Index 47 Consensus CDR3: CSARASGGEAKNIQYF V: TRBV20-1	CSARASGGEAKNIQYF, CSARAGGGEAKNIQYF	V: TRBV20-1*01 Species: M.tuberculosis Protein: Rv3874 Epitope: AAVVRFQEAAANKQKQ
Index 12 Consensus CDR3: CASSLIENTEAFF V: TRBV12-4	CASSLIENTEAFF	V: TRBV12-3*01 Species: M.tuberculosis Protein: Rv3804c Epitope: PSPSMGRDIKVQFQS

B

Supplementary Figure 9: Published Mtb reactivity of class II-restricted metaclones.

A. CDR3 sequences included in class II-restricted metaclones (n=1392) were compared against CDR3 β sequences annotated in VDJdb with MHC class "MHCII", and antigen species "M.tuberculosis" or "Mtb" (n=30). The middle column shows the matching CDR3; the left-hand column shows the description of the metaclone that the CDR3 is a member of; the right-hand column shows VDJdb annotations for the CDR3. **B.** Adjacency graphs and sequence logos for metaclones with Mtb CDR3 matches in VDJdb, and for metaclones that are most abundant (matching 47 out of 21,212 cells with β -chain TCR data) or most public (found in 14 out of 70 participants) in single cell TCRseq data from Mtb-reactive T cells (Musvosvi et al., 2023). Each node in the adjacency graph represents a single TCR stratified by colour into distinct donors from the Day 7 TST discovery dataset.

Table 3 Top 10 public metaclones in down-sampled D7 TST dataset

Index	Publicity	Consensus CDR3aa	V gene usage	Lead HLA association (odds ratio, p-value)
8	82/128	CSARVGGNTGELFF	TRBV20-1	DRB1*15 (10.2, 1.84E-09)
76	78/128	CSAGGLAGNEQFF	TRBV20-1	DQA1*01_DQB1*05 (8.6, 9.30E-06)
13	69/128	CASSLGSVSYEQYF	TRBV7-9	DRB1*15 (11.3, 3.55E-09)
21	64/128	CSARDLGLAEETQYF	TRBV20-1	DRB1*04 (11.3, 4.41E-08)
39	60/128	CSVGETQYF	TRBV29-1	DQA1*05_DQB1*03 (9.1, 4.63E-07)
33	56/128	CSARAGYGYTF	TRBV20-1	DRB1*10 (44.0, 2.61E-07)
91	50/128	CASSLEGETQYF	TRBV7-9	DRB1*11 (6.2, 2.17E-05)
6	49/128	CASSRGAQTYEQYF	TRBV18	DPB1*04 (27.5, 8.91E-10)
100	47/128	CSARGQGNEQFF	TRBV20-1	DRB1*13 (7.9, 2.73E-05)
97	43/128	CASSPGRETQYF	TRBV6-6 TRBV6-5 TRBV6-9	DRB1*10 (26.8, 2.39E-05)

15. Several publications have reported responses by donor-unrestricted T (DURT) cells to Mtb. The dataset generated in this paper allows analysis of whether any of the responding T cells in the Mtb-specific subset (infected vs peptide-pulsed APCs) include TCR-alpha sequences that match those of canonical MR1-restricted T cells, or CD1B, CD1c and CD1d-restricted T cells. I would expect low enrichment at both the d2 and d7 time points (compared to PBMCs) since intradermal PPD was used.

As per the reviewer's suggestion, we include an additional supplementary figure (Supplementary Figure 5), showing percentage of TCR-alpha sequences in blood and TST that match the gene usage described for MAIT, GEM and iNKT cells. There is no enrichment of these TCR sequences in the TST compared to blood, neither on day 2 nor on day 7. Instead, MAIT-associated TCR-alpha sequences are decreased in the day 2 TST compared to blood and further decreased in the day 7 TST, and iNKT-associated TCR-alpha sequences are also depleted in the day 7 TST compared to blood.

We have amended the text to reflect this additional analysis.

Methods lines 489-491: "MAIT TCR enrichment was assessed based on their canonical TCR α gene usage as sequences containing TRAV1-2, paired with TRAJ12, TRAJ20 or TRAJ33; iNKT TCRs were identified as TCRs containing TRAV10 paired with TRAJ18; and GEM TCRs were identified as TCRs containing TRAV1-2 paired with TRAJ9⁴⁷."

Results lines 150-153: "Day 7 TSTs showed a statistically significant reduction in the relative frequencies of both CMV and EBV-reactive CDR3 sequences compared to blood and day 2 TSTs, consistent with larger clonal expansions of Mtb-reactive sequences. These clonal expansions were not explained by donor-unrestricted T cell responses to Mtb, since day 7 TSTs showed a significant reduction in the relative frequencies of TCR α sequences that match the gene usage of MAIT or iNKT cells, compared to day 2 TSTs and/or blood (Supplementary Figure 5)."

Supplementary Figure 5: TCR alpha sequences associated with donor-unrestricted T cells are not enriched in the TST. Percentage of alpha chain sequences associated with mucosal-associated invariant T (MAIT), invariant natural killer T (iNKT) and germline-encoded mycolyl lipid-reactive (GEM) T cells in blood, day 2 and day 7 TST TCR repertoires. Abundance was calculated in full repertoires (n=20 Blood, n=17 Day 2 TST, n=165 Day 7 TST) and after individual alpha chain bulk TCR repertoires were down-sampled to 16,000 total TCRs (n=20 Blood, n=16 Day 2 TST, n=119 Day 7 TST). Data are shown as boxplots depicting median and interquartile range (IQR), while outlier data points (more than 1.5*IQR beyond the box hinges) are shown as dots. Statistical significance was assessed with Wilcoxon tests and corrected for multiple testing (ns FDR>0.05, * FDR<0.05, ** FDR<0.01, **** FDR<0.0001).

16. In line 258, the context of the "metaclones" is unclear. Are the authors referring to Mtb-specific TCRs?

We have amended the term "metaclones" in this sentence to "putative Mtb-reactive metaclones" (line 274). We believe it is already clear that our metaclones are defined by analysis of bulk TCR beta-chain data in day 7 TSTs, but are happy to make additional amendments if there are perceived to be further ambiguities.

17. The authors frequently refer to public and private metaclones (especially lines 253-275), public T cell responses, and TCRs throughout the manuscript without defining this terminology. Perhaps it simply needs to be made consistent throughout the manuscript and clearly defined in the Results Section. Grouping algorithms are used to group what would be considered "private" TCRs (unique to individuals) that contain the same central motifs as other private TCRs, and therefore usually recognize the same antigens in the context of the same MHC molecules. However, the TCRs, themselves, are not necessarily public.

We have reviewed the manuscript carefully. There are no instances of "private metacones". We have amended the Results text to make the distinction clearer between PPD-reactive CDR3 sequences that were defined in ex vivo stimulation experiments of paired PBMC (and therefore considered 'private'), and Mtb-reactive metaclones that were defined by clustering similar TCR sequences from day 7 TSTs across donors (and therefore considered 'public').

18. It is not clear what is meant by the statement in lines 246-247. If broader metaclones were not enriched, then why would it be more likely that the more specific / less broad metaclones would be?

The reviewer raises an important question for clarification. Enrichment in this analysis is a function of specificity. We have amended the text to make this point explicitly.

Lines 245-248: “To confirm that our day 7 TST derived, class II associated T cell metaclones represented public Mtb-reactive T cell responses, we calculated their enrichment in independent TCR sequencing data derived from people with TB compared to other diseases, or at the site of TB disease compared to blood. Enrichment in this analysis suggests specificity for Mtb-reactive TCRs.”

In the paragraph referenced by the reviewer, we first observe that Metaclonotypist-clustered TCRs are enriched to similar extent compared to GLIPH2-clustered CDR3s despite including a larger of proportion of day 7 TST TCRs. Since not all TCRs present in the day 7 TSTs are Mtb-reactive, the increased sensitivity of Metaclonotypist could have resulted in also clustering non-PPD-reactive TCRs. However, the comparable enrichment in independent TB-associated datasets suggests that this is not the case. Secondly, we observe that the short CDR3 amino acid motifs used by GLIPH2 to describe its clusters are not substantially enriched in TB-associated TCR sequencing datasets, indicating a lack of specificity. These short motifs capture a common sequence pattern of the included CDR3s. However, while the included CDR3s themselves are enriched in TB-associated datasets, the short motifs alone are not enriched and thus cannot be considered specific to Mtb-reactive TCRs.

19. In lines 270-272 the authors state, “Despite the dominance of private TCRs at individual level, evaluation of the cumulative publicity of metaclones ranked by their publicity suggests that as few as 10 metaclones are sufficient to identify Mtb-T cell reactivity in the entire study population”. It is not clear to me what this is intended to convey. All individuals would be expected to have mostly private and some public TCRs in any immune response. Do the authors suggest a threshold for the number of metaclones (e.g. 10) present in a PPD response that would determine whether that response was Mtb-specific? Is there reason to think any of the responses do not contain at least some Mtb-specific TCRs despite the fact that all participants were reportedly IGRA+?”

We thank the reviewer for requesting this additional clarification. The important point is that metaclones potentially resolve T cell reactivity at the level of specific peptide-MHC complexes. Therefore, despite the huge repertoire of potential peptide sequences that could be derived from Mtb, as few as 10 specific peptides are sufficient to identify an Mtb-reactive response in the whole study population. This interpretation was elaborated further in our Discussion (lines 312-317), along with the potential implications of this finding to investigate T cell correlates of clinical outcome and inform next generation vaccine design based on specific protective epitopes in place of protein antigens (lines 392-396).

Reviewer #3 (Remarks to the Author)

20. In this manuscript Turner et al. take a unique approach to determine T cell recruitment and clonality into a temporal antigen challenge site. Analysis at two timepoints (day 2 and 7) following antigen exposure provides beneficial insights into the timing and potential kinetics of T cell recruitment. Turner et al. found initial recruitment during the peak inflammatory response was predominantly non-antigen specific, as determined by no enrichment for Mtb-specific CDR3s compared to blood. These T cells were then replaced by day 7 with predominantly expanded Mtb-reactive T cells. Turner et al. developed a modular computational pipeline to identify public alpha or beta TCR sequences, establishing of a catalogue of public Mtb-specific HLA-restricted T cell sequences. While the majority of Mtb-reactive T cells were private, computational analysis revealed 10 metaclones (single chain only), suggesting population-level immunodominance of Mtb-specific responses.

We thank the reviewer for this summary.

21. All analysis seems to be performed in samples from healthy individuals with evidence of peripheral blood Mtb-reactive T cells. Do the authors believe the preexisting Mtb-reactive T cells in the peripheral blood were naïve T cells or the result of prior exposure?”

It is well established that PPD responses in peripheral blood are dependent on memory T cells. This is the basis of diagnostic use of interferon gamma release assays.

Our data do not allow us to directly distinguish between TCRs from naïve and memory T cells. In figure panels 3C-D, we find that the set of private PPD-reactive CDR3s is present in significantly greater proportion amongst blood TCR sequences with a count >1, which are enriched for memory T cells (median 7.2% for count >1 vs. 4.2 % for count >0 of all CDR3s, and median 6.8% for count >1 vs. 2.7% for count >0 of unique CDR3s). This suggests that the pre-existing Mtb-reactive T cells were predominantly expanded and thus the result of prior exposure. However, we cannot exclude that naïve Mtb-reactive T cells may also contribute to the set of in vitro defined PPD-reactive T cells to a small extent.

We have added this observation to the Results description (lines 180-183):

“Both in blood and the day 2 TST, ex vivo PPD-expanded CDR3s were present in significantly greater proportion amongst TCR sequences with a count >1 (Figure 3C-D, Supplementary Figure 7C-H), suggesting that the PPD-reactive CDR3s were predominantly expanded, as would be expected for memory T cells²⁶.”

22. While examining T cell responses at the site of antigen challenge over peripheral blood is commendable, it is worth noting that this is not the site of infection in TB. The authors should acknowledge that recruitment of T cell into the skin during a controlled antigenic challenge may be remarkably different to recruitment of T cells into sites like the lung during infection.

The reviewer raises an important potential limitation. Hence it was particularly important for us to confirm enrichment of Mtb-reactive metaclones derived from the TST in TCR sequencing data from the site of TB lung disease compared to lung cancer, and compared to the general circulating repertoire (Figure 5A) in patients with TB.

We have amended the Discussion as follows (lines 306-311):

“T cell metaclones derived from the day 7 TST reveal public T cell responses that are highly enriched in multiple sources of TCR sequence data from the blood and lung tissue of patients with TB compared to other diseases, and at the site of pulmonary TB compared to blood. The enrichment of TST metaclones in the TB lung suggests that T cells recruited into PPD-challenged skin have clinical relevance and supports the use of the TST as experimental human challenge model for immune responses in TB.”

23. Figure 3B, does each column across the x-axis represent one participant?

The reviewer is correct in their interpretation, and we have amended the figure legend for clarification:

“(B) Heatmap of unique ex vivo PPD-reactive β -chain CDR3s (clustered by Ward D2 linkage). Each column across the x-axis represents one participant.”

24. Which figure shows the lower enrichment of the full set of day 7 TST TCR clones compared to the metaclones (page 8, line 249)?

This is shown in Figure 5A, comparing grey datapoints (Discovery TST CDR3s) to blue datapoints (Metaclonotypist regex). We have amended the Results description for clarification (lines 262-266):

“Interestingly, compared to metaclones, we found substantially lower enrichment of the full discovery set of expanded day 7 TST β TCRs (Figure 5A; grey vs. blue), suggesting that a substantial proportion of day 7 TST TCR clones may not be specific to Mtb, and that identification of metaclones significantly improves antigen-agnostic enrichment of the Mtb-reactive T cell response.”

25. One limitation of this bulk sequencing approach, which the authors acknowledge, is the lack of paired alpha/beta-chain data, with the public metaclones identified being restricted to a beta-chain only. I recommended the authors are careful in the language about generalizing immunodominance to specific peptides (line 292) as the metaclones identified are only beta-chains and therefore insufficient to pursue reverse epitope discovery.

We agree with the reviewer and discuss this limitation explicitly in lines 388-396.

We have amended the discussion (lines 312-317) to reflect the reviewer’s concern:

“Nonetheless, the cumulative publicity of the most public metaclones indicates striking population-level coverage of Mtb reactivity. Although metaclones are defined by their β -chain motifs only, we hypothesize that this reflects immunodominance of specific Mtb epitopes. This finding extends previous studies that report generalisable antigenic immunodominance at the protein level^{12–14} by providing a scalable antigen-agnostic approach to potentially identifying generalisable immunodominance at the level of specific peptides.”

26. The work appears to be novel and supports the conclusions of the authors, providing interesting data regarding the timing, clonality and publicness of T cell recruitment into an antigenic site. However, without paired alpha/beta TCR data, the applicability of these findings to disease stratification and epitope discovery appears to be limited.

We thank the reviewer for highlighting the novelty and interest of our results. As regards the identification of paired α/β TCRs for these metaclones, we agree that this represents an important next step, which we also discuss in lines 390–396.

While a comprehensive identification of paired chain motifs is beyond the scope of the current study, our preliminary results demonstrate the feasibility of linking metaclones to paired alpha/beta TCRs. For 87 of the 177 class II restricted metaclones, we identified CDR3 beta motif matches in single-cell TB-lung data (~10k unique clones) or the single cell data reported by Musvosvi et al. (~17k unique clones). Therefore, even at the current scale of data, many metaclones can readily be extended to paired alpha/beta TCRs, thus enabling reverse epitope discovery. In ongoing work, we are generating larger single-cell datasets to further increase recovery of paired α/β TCRs for our metaclones, which will be the subject of a future publication.

Taken together, these findings suggest that the lack of paired chain information, while a current limitation, does not substantially impact the applicability of our findings.

We confirm that our manuscript meets all of the stipulations detailed above.

Response to reviewers

As requested, we have provided point by point responses to the reviewers' comments and indicated the changes we have made below.

Reviewer #1

1. *Thanks to the authors for replying to my comments so extensively, including additional analysis. I think these extra data have improved the manuscript and clarified my initial concerns on TCR numbers.*

We thank the reviewer for their supportive comments.

Reviewer #2

2. *All of my comments have been addressed.*

We thank the reviewer for their supportive comments.

Reviewer #3

3. *Thank you for the revised manuscript and detailed response letter. While the majority of my comments have been addressed and the manuscript provides new insights into TCR recruitment to a temporal antigen challenge site, without paired TCR data or experimental validation of HLA restriction/public clones, the applicability of these findings as a resource for biomarker discovery and reverse epitope discovery is limited and the work likely does not significantly advance the field.*

We thank the reviewer for their supportive comments. We acknowledge the reviewer's comments about the limitations of the chain analyses used in the present manuscript, which we have discussed explicitly (lines 390-396) and is the focus of future work to collate paired TCR data for TST metaclones to experimentally validate their HLA restriction and antigen reactivity.

4. *Outside of this, my only other suggestion is to include the additional details about the participant cohort in the results section of the paper (line 90). The authors response to Reviewer #1, that they are healthy volunteers with known exposures, was a helpful qualification and I believe the manuscript will be strengthened with more upfront inclusion of this information.*

As suggested by the reviewer, we have elaborated on the context in which immunological memory to *Mtb* was identified in study participants, arising from occupational health, contact tracing and new migrant screening programmes (line 91-92).